# Hydrogen-bonded organic framework biomimetic entrapment allowing non-native biocatalytic activity in enzyme

Guosheng Chen [1,4] ✉, Linjing Tong[1,4], Siming Huang [2], Shuyao Huang[3], Fang Zhu[1] & Gangfeng Ouyang [1,3] ✉

Nature programs the structural folding of an enzyme that allows its on-demand biofunctionality; however, it is still a long-standing challenge to manually modulate an enzyme's conformation. Here, we design an exogenous hydrogen-bonded organic framework to modulate the conformation of cytochrome c, and hence allow non-native bioactivity for the enzyme. The rigid hydrogen-bonded organic framework, with net-arranged carboxylate inner cage, is in situ installed onto the native cytochrome c. The resultant hydrogen-bonded nano-biointerface changes the conformation to a previously not achieved catalase-like species within the reported cytochrome c-porous organic framework systems. In addition, the preserved hydrogen-bonded organic framework can stabilize the encapsulated enzyme and its channel-like pores also guarantee the free entrance of catalytic substrates. This work describes a conceptual nanotechnology for manoeuvring the flexible conformations of an enzyme, and also highlights the advantages of artificial hydrogen-bonded scaffolds to modulate enzyme activity.

Nature utilizes enzymes to perform a myriad of biological functions unmatched by synthetic counterparts[1,2]. One of the most impressive examples is found in the heme enzymes, wherein the heme units are immobilized in diverse protein scaffolds to carry out specific tasks, including substrate oxidation[3,4], electron transfer[5,6], sensing[7], metal ion storage[8], and transport[9]. For instance, cytochrome c (Cyt c), in which the heme is covalently bonded with the protein scaffold through two disulfide bonds and axially coordinated by histidine (H18) and methionine (M80), is a highly stable heme protein that severs as a component of the electron transport chain in mitochondria[10]. While in the oxidoreductase such as catalase (CAT) and peroxidases, the hemes are found to be immobilized by weak non-covalent interactions[11], and their Fe active sites were coordinated by only one axial amino acid. The formed high spin Fe species favors the substrate affinity and hence accelerates the biocatalysis[12]. As a result, these oxidoreductases play a crucial protective role in preventing the oxidative damage of cellular components caused by reactive oxygen species, as well as its highly reactive decomposition products[13,14]. We envy nature's ability to program enzymatic conformations in highly crowded cellular environments and afford the enzymes with on-demand bioactivities. Increasing efforts have implied that this sophisticated programming depends on a set of conserved proteins known as molecular chaperones[15], for instance, the typical GroEL-GroES chaperonin in bacteria[16] (Fig. 1a). The GroEL-GroES system is a large double-ring complex. It has a cage-like structure enabling the encapsulation of unfolded or non-native proteins, and then formed a highly hydrophilic, net-negatively-charged inner wall to modulate the protein folding[17]. This inspires

[1]MOE Key Laboratory of Bioinorganic and Synthetic Chemistry, School of Chemistry, Sun Yat-sen University, Guangzhou 510275, China. [2]Guangzhou Municipal and Guangdong Provincial Key Laboratory of Molecular Target & Clinical Pharmacology, the NMPA and State Key Laboratory of Respiratory Disease, School of Pharmaceutical Sciences and the Fifth Affiliated Hospital, Guangzhou Medical University, Guangzhou 511436, China. [3]Instrumental Analysis and Research Center, Sun Yat-sen University, Guangzhou 510275, China. [4]These authors contributed equally: Guosheng Chen, Linjing Tong.
✉e-mail: chengsh39@mail.sysu.edu.cn; cesoygf@mail.sysu.edu.cn

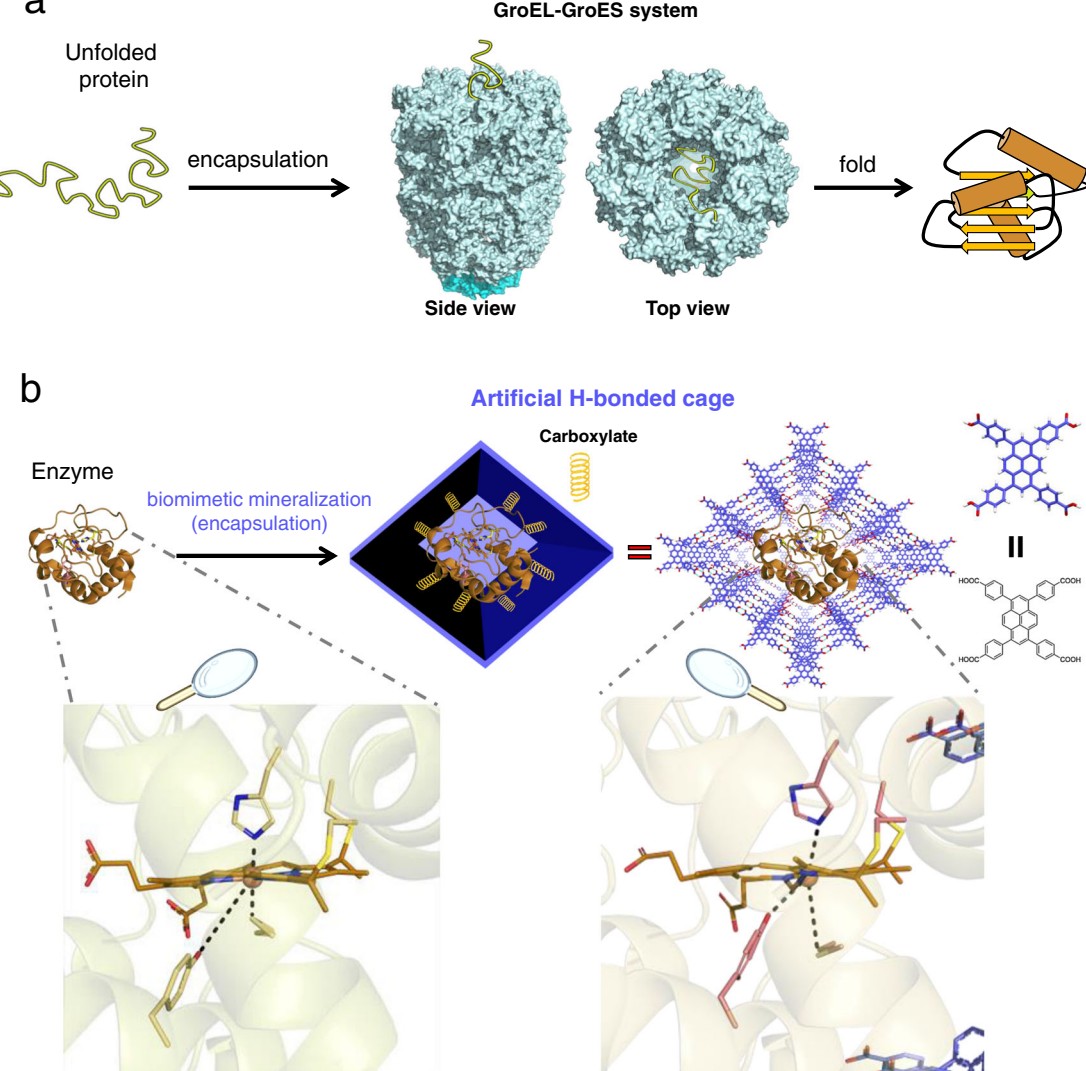

**Fig. 1 | Modulating the conformation of an enzyme by H-bonded cage encapsulation. a** Schematic illustration of the protein folding in the typical GroEL-GroES chaperonin system (PDB: 1pf9)[16]. **b** Schematic illustration of the designed H-bonded nanocage for modulating the conformation of an enzyme, Cyt c (PDB: 6k9i). The colors used in HOF-101 are: red for O atom; blue for C atom; white for H atom. The colors used in heme of Cyt c are: brown or pink for C atom; blue for N atom; yellow for S atom; red for O atom; the central Fe ion is highlighted as an orange ball.

the scientists to utilize synthetic nanocages for modulating the conformation of enzyme, yet, still remains a long-standing challenge.

Over the past decade, substantial efforts have been devoted to encapsulating enzymes within porous frameworks from metal-organic frameworks (MOFs)[18–21], covalent organic frameworks (COFs)[22] to newly emerging hydrogen-bonded organic frameworks (HOFs)[23–26]. Viewing from the linkage topologies, HOFs are constructed by the weakly intermolecular interactions of hydrogen bonds. This distinct linkage topology endows HOF with a much milder crystallization process, wherein the high pressure and temperature or strong acidity that are usually involved in MOFs or COFs synthesis[27–31] are not required. Considering this attractive attribute, in situ encapsulation of enzymes into HOFs is highly desirable because the mild crystallization conditions can circumvent the denaturation risk of enzyme during the assembly. For example, Doonan, Falcaro and others have reported the possibility of in situ encapsulation of enzymes into the water-stable, amidinium···carboxylate-based HOFs[24–26]. The confined HOFs microenvironment can stabilize the enzyme, and hence enhances its durability. However, to our best knowledge, the current encapsulation technology is still unable to control over the enzyme's conformation that allows the non-native bioactivity.

Here, we show the possibility of utilizing a HOF to modulate the enzyme's conformation, and hence offer the enzyme different biofunctionality (Fig. 1b). A rigid HOF, with net-carboxylate-arranged defective cages, is in situ installed onto the native Cyt c (from *Equus caballus* heart) surface via biomimetic mineralization. Experimental data and molecular dynamics simulation indicate that the encapsulated Cyt c changes its native conformation due to the H-boned nano-biointerface between Cyt c and the artificial HOF cage. Especially, the low spin, hexa-coordinated Cyt c heme is modulated into a CAT-like high spin, five-coordinated species that has previously not been achieved in the reported Cyt c-porous organic framework systems. In addition, the hydrophobic pocket, where the active heme is almost completely buried into, is also opened by the interfacial interaction of the HOF, which favors the biocatalysis. The designed HOF has long-range ordered channels, not only stabilizing the encapsulated enzymes, but also enabling the substrate diffusion. Our Cyt c@HOF nanosystem can carry out the CAT-like catalytic function in different harsh conditions while the free enzyme will be denatured, and displays

significant enhancements on stability and reusability. This work showcases the conception of modulating an enzyme's conformation by a HOF.

## Results

### Installing the HOF onto Cyt c

For modulating the enzyme's conformation, we require the artificial HOF to have size-matched cages for enzyme encapsulation, and the nano-biointerface between the enzyme and the cages should be designed. Hydrogen-bonded framework-101 (HOF-101)[32,33] is an ideal choice because: 1) its mild assembly condition, and the structural stability originated from its strong layer-by-layer π⋯π stacking structure, and 2) the periodic carboxylate networks in HOF-101 may favor the formation of nano-biointerface because of the potential hydrogen-bonded interactions between the surface residues of enzymes and the carboxylate networks. Herein, the encapsulation of Cyt c into HOF-101 was realized by a biomimetic mineralization process[18,23], wherein the enzyme triggered the HOF-101 nucleation around its surfaces and a net-carboxylate-arranged defective cage was formed for accommodating the enzyme (Supplementary Fig. 1). We envisage that such net-carboxylate-arranged cage may mimic the chaperone cage, because the net-carboxylate arranged in the inner wall of the defective cage results in a highly hydrophilic, net-negatively-charged microenvironment, as like the ones of chaperone cage after protein encapsulation[17].

When the HOF-101 was installed onto Cyt c, the apparent color of the material turned from yellow to brownness (Fig. 2a), because of the encapsulation of Cyt c. The standard Bradford assay (Supplementary Fig. 2a) gave a ca. 39 wt% Cyt c content in the nanosystem (Supplementary Table 1). Such a high enzyme loading was also supported by the inductively coupled plasma-mass spectra (ICP-MS) measurement (Supplementary Fig. 2b, c), wherein 0.156 wt% Fe in average was detected in the biocomposites (equaled to 36.2 wt% Cyt c in average).

We next designed a surface-adsorption experiment to confirm that Cyt c could not be surface-adsorbed by HOF-101, as evidenced by the UV-Vis and Fourier transform-infrared spectrum (FT-IR) data (details seen in Supplementary Fig. 3). The closer insight into the significant reduction of $N_2$ adsorption amount of Cyt c@HOFs compared to parent HOF-101 (Supplementary Fig. 4) also verified that the Cyt c was indeed encapsulated, rather than surface-adsorbed onto HOF-101. Such HOF-101-encapsulated Cyt c (Cyt c@HOF-101) nanostructure was further supported by other characterizations, including powder X-ray diffraction (Supplementary Fig. 5), thermogravimetric analysis (Supplementary Fig. 6), and small-angle X-ray scattering (Supplementary Fig. 7).

The atomic-level structure of Cyt c@HOF-101 was profiled by low-electron-dose cryoelectron microscopy (cryo-EM, Fig. 2b). The long-range ordered channels with 2.0 nm width were witnessed throughout the biocomposite (Fig. 2b and Supplementary Figs. 8 and 9), which should favor the catalytic substrates diffusion and products transportation. The typical atomic-level information (Fig. 2c), viewed from the [01$\bar{1}$] projection, was clearly presented by the inverse fast Fourier transformation (IFFT) image. The periodically arranged bright spots were assigned to the cavities, which were formed by the intermolecular H-bonded assembly and layer-by-layer π⋯π stacking of the organic linkers. Such information seen by the cryo-EM was well in agreement with the crystallographic structure of HOF-101, wherein the H-bond connected layers were stacked by A-A stacking, with a ca. 0.37 nm interlamellar spacing (Fig. 2d).

The spatial distribution of Cyt c was profiled by confocal laser scanning microscopy (CLSM) experiments, wherein the Cyt c was labelled by rhodamine b (RhB, a red fluorescence dye). The red fluorescence completely overlaid with the frameworks (Fig. 2e and Supplementary Fig. 10), suggesting the uniformly spatial distribution of Cyt c. These results indicated the exogenous HOF was installed onto Cyt c as we expected (Fig. 2f).

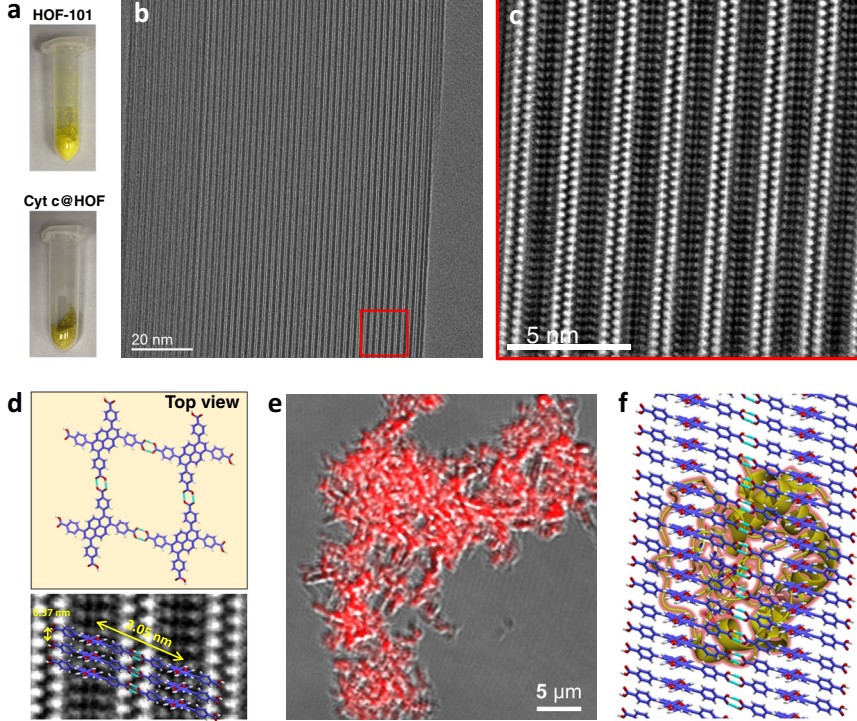

**Fig. 2 | Structural profile of Cyt c@HOFs. a** The digital photographs of the HOF-101 and Cyt c@HOFs. **b** Cryo-EM image of the synthesized Cyt c@HOF-101 when viewing from [01$\bar{1}$] projection. **c** The IFFT image of the fast Fourier transformation of the red area in **b** shows the atomic-level information of Cyt c@HOF-101. **d** Detailed structural analysis of the Cyt c@HOF-101. **e** CLSM shows the spatial distribution of Cyt c (labelled by red dyes) within the framework, and **f** the schematic presentation of Cyt c@HOF-101. The colors used in HOF-101 are: red for O atom; blue for C atom; white for H atom.

## The conformational structure and functional activity

The interfacial interaction between Cyt c and the installed HOF was then examined. The solid-state nuclear magnetic resonance (ssNMR) spectra of Cyt c@HOF-101 were different from those of the free Cyt c and the physical mixing of Cyt c and HOF-101 (Supplementary Fig. 11). Especially, the chemical shifts of [1]H ssNMR ranged from 14 to 15 ppm and [13]C ssNMR ranged from 170 to 175 ppm, assigned to the −COOH group of HOF-101, was observed to be shifted in Cyt c@HOF-101. It suggested the additional interaction between HOF-101 and Cyt c in the Cyt c@HOF-101 nanosystem. Furthermore, the closer examination by FT-IR spectra found that, the typical amide I and amide II bands of native Cyt c (ascribed to the C=O stretching and N-H bending[34], respectively) were perturbed after encapsulation by HOF-101, further supporting the interfacial interaction between Cyt c and HOF-101 (Supplementary Fig. 12).

The structural change of Cyt c was then examined. We firstly utilized circular dichroism (CD) spectrum to verify that the native secondary structure of Cyt c was well maintained after assembly (Supplementary Fig. 13). The microenvironments of the active center of Cyt c were profiled by UV-visible diffuse reflectance spectroscopy (UV-Vis DRS, Fig. 3a), a sensitive means for heme coordination study. The distinct UV-Vis DRS peaks at 520 nm and 550 nm are the characteristic peaks of hexa-coordinated low-spin heme in native Cyt c[35,36], while the peaks at 696 nm is assigned to the characteristic electron transfer peak of heme-M80 residue[37,38]. Markedly, these three typical absorption bands completely missed in Cyt c@HOF-101, and a typical peak of high spin heme at 625 nm[39,40] emerged. It indicated the axially coordinated M80 was removed and a non-native high spin, five-coordinated heme specie was formed. Importantly, the UV-Vis DRS profile of Cyt c@HOF-101 was similar to the native CAT (details were

discussed in the molecular dynamics simulations below). Such HOF-101-induced coordination environment change of heme was also verified by ultralow temperature (5 K) electron paramagnetic resonance (EPR) spectroscopy, wherein the high spin ferric hemes[41] with g-values of [5.93, 5.66, 1.99] were retained and the low spin ferric hemes[42] with g-values of $g_z$ (2.93) and $g_y$ (2.26) had almost completely disappeared (Fig. 3b). These data suggested the native Cyt c conformation indeed changed after the encapsulation by HOF, and the formed Cyt c@HOF-101 has CAT-like biofunctionality (Fig. 3c), while retained its intrinsic peroxidase activity (Supplementary Fig. 14).

Cyt c@HOF-101 was capable of catalyzing the decomposition of $H_2O_2$ into $H_2O$ and $O_2$ (experimental details seen in Supplementary Table 2), even though its activity still could not be comparable with the native CAT. This catalytic activities depended on the substrate concentrations (Fig. 3d), and the catalytic kinetics was accorded well with the Michaelis-Menten equation of single substrate enzymatic reaction (Supplementary Fig. 15). We firstly verified the mild liquid phase (deionized water/DMF = 9/1, v/v) used in our synthetic process, could not disturb the native conformation of Cyt c, as evidenced by the intact adsorption bands in UV-Vis spectra of the resultant Cyt c (Supplementary Fig. 16a). In addition, no distinct CAT-like activity was recorded when Cyt c was directly exposed to this condition (Supplementary Fig. 16b). To elucidate the key role of the net-carboxylate-arranged nanocage for the directional modulation of enzyme's conformation, we also synthesized other prosperous Cyt c-framework nanocomposites through biomimetic mineralization (Fig. 4a), including Cyt c@ZIF-8[18,43–45] (Supplementary Fig. 17) and Cyt c@ZIF-90[46] (Supplementary Fig. 18). The UV-Vis DRS profiles suggested that these MOF encapsulation strategies were unable to modulate the conformation of Cyt c (Fig. 4b). Another recently reported Cyt c@NU-1000

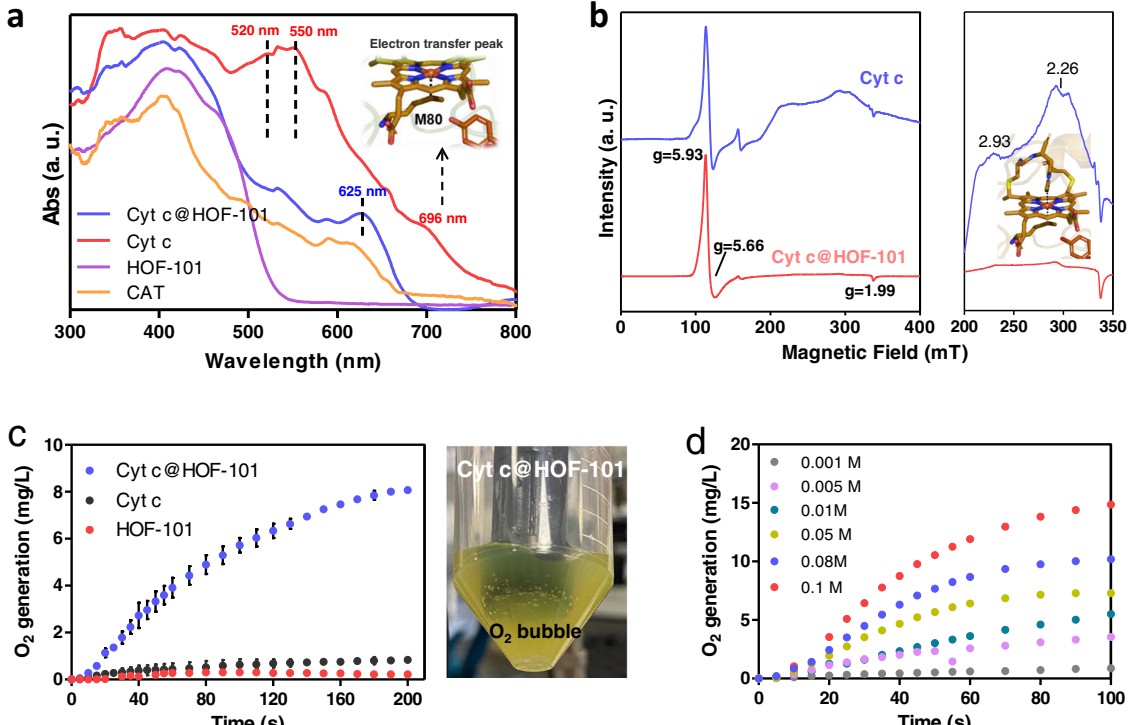

**Fig. 3 | The conformation change and CAT-like biofunctionality. a** Normalized UV-Vis DRS of Cyt c@HOF-101, Cyt c, HOF-101 and CAT. **b** The 0–400 mT and the zoomed 200–350 mT regions of the EPR spectra of free Cyt c and Cyt c@HOF-101 collected at 5 K. The colors used in heme of Cyt c in **a**, **b** are: brown for C atom; blue for N atom; red for O atom; yellow for S atom; the central Fe ion is highlighted as an orange ball. **c** The capacity of $O_2$ generation by $H_2O_2$ decomposition of Cyt c@HOF-101, Cyt c and HOF-101, and the digital photograph shows the $O_2$ bubble generation

using Cyt c@HOF-101 catalyst. All tests were carried out in the Tris buffer (pH 7.5, 50 mM). Cyt c dosage in each trial including free Cyt c and Cyt c@HOF-101 nanosystem was kept at 0.1 mg/mL, and the final $H_2O_2$ concentration was 10 mM. Error bars (SD) are presented, SD = Standard Deviation (n = 3). Data are presented as mean values ± SD. **d** The $H_2O_2$-dependent catalytic kinetic curves of Cyt c@HOF-101 catalyst. All tests were carried out at Tris buffer (pH 7.5, 50 mM). Cyt c dosage was kept at 0.1 mg/mL.

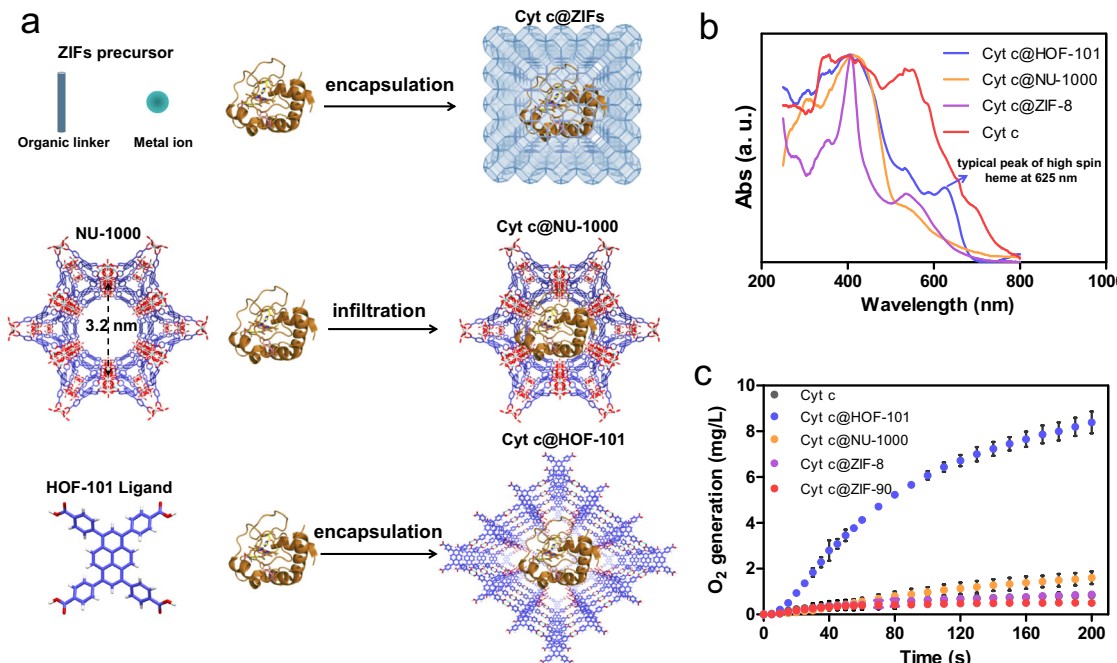

**Fig. 4 | The key role of the designed net-carboxylate-arranged cage for conformational modulation. a** Schematic illustration of the synthesis of Cyt c@ZIFs, Cyt c@NU-1000 and Cyt c@HOF-101. The colors used in NU-1000 are: red for O atom; blue for C atom; light green for Zr atom; H atom is removed for clarity. The colors used in HOF-101 are: red for O atom; blue for C atom; white for H atom. The normalized UV–Vis DRS (**b**) and CAT-like bioactivities of Cyt c@ZIFs, Cyt c@NU-1000 and Cyt c@HOF-101 (**c**). The dosage of Cyt c in each trial including free Cyt c and different Cyt c nanosystem groups were kept the same (0.1 mg/mL). Error bars (SD) are presented, SD = Standard Deviation (n = 3). Data are presented as mean values ± SD.

nanosystem, wherein the Cyt c was post-infiltrated into the cavity of an intact NU-1000 crystal (Fig. 4a and Supplementary Fig. 19), has shown the possibility of disturbing the heme structure of Cyt c[47]. However, this mild post-infiltration strategy was also unable to directly modulate the Cyt c heme into a CAT-like species, as seen in the UV-Vis DRS profile (Fig. 4b). Consequently, the formed Cyt c@NU-1000 also displayed very limited CAT-like activity (Fig. 4c). These results together clarified the superiority of our H-boned scaffold, which has net-carboxylate-arranged cages, for the conformational modulation of the enzyme, achieving non-native biofunctionality.

## Insight into the conformation change by computational simulation

Further insight into the conformational structure of Cyt c@HOF-101 was examined by all-atom explicit solvent molecular dynamics (MD) simulations (details seen in Supplementary Discussion and Supplementary Figs. 20–22). The flexible turns with a mass of $NH_2$-rich lysine (such as K87, K86, K72, and K73, region A in Fig. 5a), as well as the short helices with lysine (K8 and K27) and other polar residues of glutamine (Q16), valine (V11) and threonine (T28) of the encapsulated Cyt c (region B in Fig. 5a), were observed to strongly interact with the net-carboxylate arranged in the inner wall of the defective cage via H-bond. The traction by the H-bond between surface residues of Cyt c and HOF-101 resulted in the global conformation change of Cyt c in Cyt c@HOFs, while the conformation of free Cyt c under the same simulating conditions remained unchanged (Supplementary Fig. 23). This H-bonded interface was well in agreement with the aforementioned [1]H ssNMR (Supplementary Fig. 11) and FT-IR data (Supplementary Fig. 12).

We then tracked the time-depended microenvironment change of the heme, in terms of the spatial position change of three important amino acids, including the axially coordinated histidine (H18) and methionine (M80) and the distal tyrosine (Y67) (Fig. 5b and Supplementary Figs. 24 and 25). Viewing from region A in Fig. 5a, the dual traction of flexible turn 69–87 of Cyt c by the formed H-bonded interface led to the departure of M80 from the Fe active center of Cyt c

(from 2.1 Å to 4.1 Å). The distance of 4.1 Å meant that the axially coordinated M80 turned into the uncoordinated state, which was also supported by the disappeared electron transfer peak of heme-M80 in UV-Vis DRS data (Fig. 3a). Interestingly, we found the distal Y67 was gradually closer to the Fe active center of Cy t c (from 4.8 Å to 2.8 Å, Supplementary Fig. 24) at the same time. When looking into the region B in Fig. 5a, the traction of short helices by the artificial H-bonded interface also caused the spatial departure of H18, and showed the fluctuation of a distance at 2.8–3.3 Å from Fe active center (Supplementary Fig. 24). Distinctly, these conformation changes resulted in a non-native high spin, five-coordinated heme active center, as confirmed in the UV-Vis DRS (Fig. 3a) and EPR spectroscopy (Fig. 3b). We also confirmed that the heme microenvironment of free Cyt c remained unchanged under the same simulating conditions (Supplementary Fig. 25). Importantly, the active center of the nanocage-encapsulated Cyt c was analogous to that of the native CAT, wherein the heme was axially coordinated by tyrosine, with the distal histidine acts as an acid-base catalyst for the $O_2$ generation by the decomposition of $H_2O_2$[48,49].

In the native Cyt c, the heme active center is buried into a flat and narrow cavity (Fig. 5c), and the computational simulation suggested that the $H_2O_2$ was hard to access the heme in the native conformation (Supplementary Fig. 26a). Amazingly, in the HOF-101-induced conformation, a larger binding cavity was shaped (Supplementary Fig. 27). This cavity was composed of amino acids arginine 38 (R38), glutamine 42 (Q42), tyrosine 48 (Y48), leucine 35 (L35), leucine 32 (L32), proline 30 (P30), aspartic acid 31 (N31), and histidine 33 (H33), of which the closed R38 amino acid at the top right of the cavity became open. It led to a ca. 6.8 Å pocket that allowed the $H_2O_2$ entrance (Fig. 5c and Supplementary Fig. 26b). Importantly, the $H_2O_2$ was capable of binding with Fe active site with ca. −4.3 kJ/mol energy (Supplementary Fig. 28a), which is a common stage in both the catalase and peroxidase reactions[3,48]. In addition, a H-bonded network of $H_2O_2$, stabilized by the polar residues within the heme pocket, was seen in Cyt c@HOF-101 (Supplementary Fig. 28b). Such H-bonded network might also

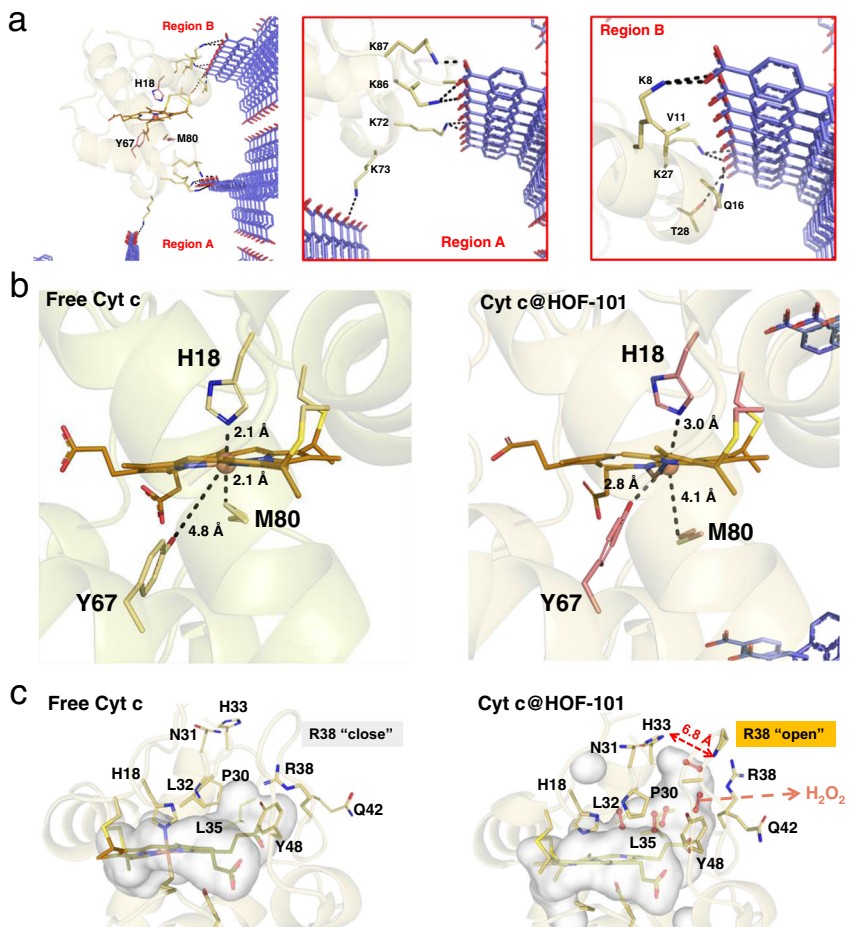

**Fig. 5 | Insight into the conformation change by computational simulation.**
**a** The typical H-bonded interactions between the Cyt c surface residues and the
−COOH groups of HOF-101. **b** The microenvironments of heme center of native Cyt
c and Cyt c@HOF-101. **c** The binding cavities of native Cyt c and Cyt c@HOF-101,
wherein the open pocket (R38) in Cyt c@HOF-101 facilitated the entrance of $H_2O_2$
into the binding cavity. The colors used in HOF-101 in **a**, **b** are: red for O atom; blue
for C atom; H atoms are removed for clarity. The colors used in heme of Cyt c in
**b**, **c** are: brown or pink (the right in b) for C atom; blue for N atom; yellow for S
atom; red for O atom; the central Fe ion is highlighted as an orange ball.

facilitate the proton transfer pathway for CAT catalysis[49]. These
simulated data gave a deep insight into the nanocage–modulated
conformation change of Cyt c, and might provide a deeper under-
standing of its CAT-like biological function.

**Catalase-like biocatalysis performance**
Among the heme enzymes, Cyt c is the only species wherein the heme
macrocycle is covalently linked to the protein skeleton by two disulfide
bonds[11]. Hence, the heme macrocycle of our Cyt c nanosystem is more
structurally stable than that in other heme enzymes, such as CAT
(Fig. 6a). Besides, the rigid artificial HOF, with ultrahigh chemical sta-
bility (Supplementary Fig. 29), can prevent the interior Cyt c from the
external stimuli. Meanwhile, the long-range ordered channels of HOF-
101 (2.0 nm width) ensure the free diffusion and transportation of
many catalytic substrates and their products, yet exclude the inter-
ferents such as large macromolecules. Such distinct nanoarchitecture
may allow our Cyt c@HOF-101 system to perform catalytic tasks in
harsh environments. It is well known that the conformation of native
CAT is highly susceptible to external stimulus. When free CAT and Cyt
c@HOF-101 were exposed in the scenarios such as non-native pH
(Fig. 6b and Supplementary Fig. 30), heating (Fig. 6c and Supple-
mentary Fig. 31), denaturing reagents (urea, hydrolase and heavy metal
ions, etc.) and organic solvents (Fig. 6d and Supplementary Fig. 32 and
Supplementary Fig. 33) environments for 30 min, respectively, the
retained CAT-like bioactivities were examined (experimental details

seen in Supplementary Table 3). Herein, the 100% retained activity of
Cyt c@HOF-101 or CAT was referred for its original bioactivity in pH = 7
deionized water at room temperature, respectively. And the mea-
surement of bioactivity conversion was based on the change of the
initial catalytic rate, which was evaluated by the slope of the kinetic
curve (Supplementary Figs. 30–33) in the initial phase from 0 to 40 s.
As seen in Fig. 6b–d, Cyt c@HOF-101 retained desirable CAT-like
bioactivity. As a comparison, the native CAT was observed to sig-
nificantly lose its bioactivity under similar treatments. Especially, the
native CAT was almost devitalized (the retained bioactivities was less
than 10%) when being operated in strong acid conditions (pH = 2), high
temperature (80–100 °C) and the environments with ethanol or $Cr^{3+}$
exposure.

To further elucidate the superiority of our Cyt c@HOF-101 system,
we also assembled the CAT@HOF-101 using the similar biominer-
alization process (Supplementary Fig. 34). The typical amide I and
amide II bands of protein appeared in the FT-IR spectra of CAT@HOF-
101, suggesting the successful encapsulation of CAT (Supplementary
Fig. 34c). In addition, the standard Bradford assay gave a ca. 44 wt%
CAT loading in CAT@HOF-101. The typical UV-Vis adsorption bands of
CAT heme were well maintained after the encapsulation (Supple-
mentary Fig. 35a), indicating that the native conformation of the heme
center was reserved in CAT@HOF-101. However, the activity of CAT
was slightly inhibited after encapsulation (Supplementary Fig. 35b),
which might be caused by the inevasible inhibition on the substrate

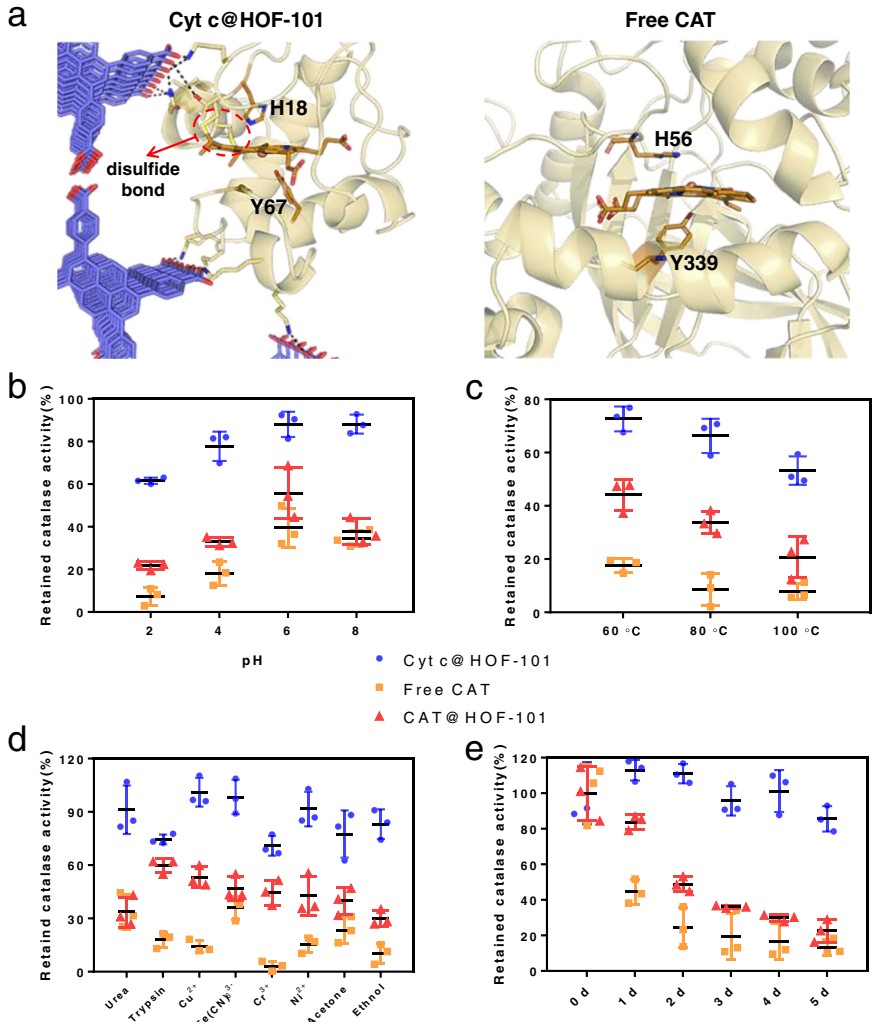

**Fig. 6 | Catalase-like biocatalysis performance. a** The structure of the heme macrocycle of our Cyt c nanosystem base on MD simulation and native CAT (PDB: 7di8). The colors used in HOF-101 are: red for O atom; blue for C atom; H atoms are removed for clarity. The colors used in heme of Cyt c or CAT are: brown for C atom; blue for N atom; red for O atom; two disulfide bonds of Cyt c are highlighted in yellow, and the central Fe ion is highlighted as an orange ball. **b–e** The retained CAT-like activities of native CAT, CAT@HOF-101 and Cyt c@HOF-101 after different conditions treatments: (**b**) Enzymes or biocomposites were exposed in deionized water with different pH at room temperature for 30 min. The final $H_2O_2$ concentration used in the activity test was 10 mM. **c** Enzymes or biocomposites were dispersed in pH = 7 deionized water, and then exposed at different temperatures for 30 min. The final $H_2O_2$ concentration used in the activity test was 10 mM.

**d** Enzymes or biocomposites were exposed in different solutions at room temperature for 30 min. Urea: 6 mol/L; Trypsin: 5 mg/mL; All of the metal ions: 10 mmol/L; Organic solvents: 80% (v/v). The final $H_2O_2$ concentration used in the activity test was 10 mM. **e** The evaluation of the storage stability of native CAT, CAT@HOF-101 and Cyt c@HOF-101 in terms of the variation of CAT-like activity. Enzymes or biocomposites were dispersed in pH = 7 deionized water, and then stood at 35 °C for different periods. The final $H_2O_2$ concentration used in the activity test was 10 mM. In **b–e**, the symbols of blue circle, orange square, red triangle represent Cyt c@HOF-101, free CAT and CAT@HOF-101, respectively. The data of 3 independent samples and calculated error bars (SD) are presented in **b–e**, SD = Standard Deviation ($n = 3$).

diffusion process by the HOF-101 shell. Such activity inhibition of CAT after encapsulation was also observed in CAT@MOFs biocomposites[46]. As shown in Fig. 6b–d, the stabilities of CAT@HOF-101 were enhanced compared to free CAT, suggesting the protecting effect of the HOF-101 shell in harsh conditions. However, after the same treatments, the activity conversions of Cyt c@HOF-101 significantly outperformed the CAT@HOF-101. These results further suggested that the extraordinary stability of our Cyt c@HOF-101 system partially resulted from the inherently structural robustness of the heme center in Cyt c.

The robust composite construction also made the cold chain-free storage or transportation of enzymes possible, where no obvious CAT-like activity of Cyt c@HOF-101 lost after storage at 35 °C for 5 d (Fig. 6e and Supplementary Fig. 36). Furthermore, the Cyt c@HOF-101 was feasible to be recycled after centrifugation, and retained ca. 70% CAT-like activity after seven times cycles (Supplementary Fig. 37).

## Discussion

Native enzymes are highly desirable owing to their efficiency, selectivity and programmability unmatched by synthetic counterparts. To gain functional activity, most enzymes must fold into defined three-dimensional structures. In this work, we pioneer to utilize the nano-technology to regulate the three-dimensional structure of Cyt c, and show the possibility of imparting Cyt c non-native functional activity through structural modulation by an artificial H-bonded scaffold. Especially, the low spin, hexa-coordinated heme of Cyt c is successfully modulated into a high spin, five-coordinated species, and the enclosed binding cavity turns into open after the conformation change. Both of these features favor the biocatalysis. Notably, this H-bonded nano-technology is easy-to-operate, and repeatable, in which different batches of synthetic Cyt c@HOF show similar CAT-like functional activities (Supplementary Fig. 38).

Our HOF nanosystem shows the possibility to modulate the conformation of an enzyme within its cages, yet, is still unable to precisely refold the native or denatured structure of an enzyme as like the chaperone function. Noted that the activity of our HOF nanosystem at present cannot be comparable with the native CAT, further works are still required to optimize the specific nano-biointerface. It also needs to point out that the exact interface between Cyt c and HOF-101 lacks unambiguous data, even though the MD simulation has given a potential biointerface process. In our nanosystem, the encapsulated enzyme resides within the HOF cage, while the folded enzymes will be released in the native chaperone cage. Considering that enzymes are highly dynamic, the remarkable advantage of our nanosystem is that the robust artificial HOF cage can stabilize the dynamic enzyme, while reserving the mesoporous channels that ensure the high accessibility of biocatalyst. This feature allows the Cyt c@HOF biosystem to perform CAT-like catalytic tasks in different harsh environments. This work highlights the advantages of the artificial H-bonded scaffold for the conformational modulation of an enzyme, since such phenomenon is seldom observed in the previous enzyme-MOFs or -COFs systems[18–22]. We believe that our findings may also provide deep insight into the structural flexibility of Cyt c under external stimuli, since increasing evidences have indicated that the conformation change of Cyt c may actually be involved in redox signaling[50], and related to many biological functions that await to be discovered[51,52].

## Methods

### In situ installation of HOF-101 onto Cyt c

The installation method was based on a biomimetic mineralization process[23]. 10 mg organic linker, H4TBAPy, was dissolved in 1 mL DMF by ultrasonic treatment, named as solution A. 5 mg Cyt c (from *Equus caballus* heart) was dissolved in 9 mL deionized water, named as solution B. Subsequently, solution B was poured into solution A rapidly under stirring. The mixed solution was stirred for 5 min, and then aged for 20 min. The obtained Cyt c@HOF-101 biocomposite was collected by centrifugation, and washed with deionized water 2 times and ethanol 1 time in sequence. For structural analysis, the biocomposite was dried under a vacuum oven at room temperature. Otherwise, the biocomposite was directly dispersed in deionized water, and stored at 4 °C.

### Cryo-EM

The cryo-EM imaging was performed on a Thermofisher Scientific Titan Krios G3i electron microscope at 300 kV[53]. In brief, the Cyt c@HOF-101 particles were dispersive in ethanol by ultrasonic treatment, and then mounted onto a TEM grid (carbon-coated grid). Herein, the specimen was manually plunge-frozen in liquid nitrogen without any blotting, and then transferred into the microscope by means of a cryo-transfer loader. The detector used was a K3 Summit direct electron detector, which was equipped with a GIF Quantum energy filter (slit width 20 eV). The used electronic dose rate was ca. 15 counts/pixel /second, and each micrograph involved a total dose rate of ca. 30 e⁻/Å². For imaging, the nominal magnification was set at 350,000, with a physical pixel size of 0.34 Å by 0.34 Å. The cryo-EM data acquisition was carried out by SerialEM software (version 3.8), and each micrograph stack consisted of 15 frames. The acquired micrograph was undergone a motion correction, which was executed by Motion-Corr2 with $2 \times 2$ binning. The non-dose-weighted sum of all frames from each movie was used for all image processing steps. DigitalMicrograph (Gatan) software (version 3.23.1518.0) was used for the analysis of the lattice spacing of the unit cells.

### All-atom explicit solvent molecular dynamics simulations

In our biomimetic mineralization process, the enzyme was confined within the defective cavity of HOF-101. We employed the setup of a defective HOF-101, which has the sized-matched defective cavity for Cyt c accommodation (details seen in Supplementary Discussion and Supplementary Fig. 20). The Cyt c@HOF-101 system with maximum clustering phase and lowest energy was selected for the further molecular dynamics simulation (details seen in Supplementary Discussion, Supplementary Fig. 21, and Supplementary Table 4). The all-atom explicit solvent molecular dynamics simulation was performed by NAMD software (version 2.15). Ff14SB force field was used in the whole enzyme, while FFGMX force field parameters were added to defective HOFs-101 to set the force field of carbon atoms and oxygen atoms with different bond angles. Then, with the Cyt c@HOF-101 system as the center, a 1 nm cubic $H_2O$ box was added, and $Cl^-$ was added to make the system electrically neutral. In the overall simulation process, 8370 kJ/mol limiting force was added to maintain the stability of the HOF-101 structure. Molecular dynamics simulation procedure is as follows: (1) Two-step energy minimization. The Cyt c was firstly restricted, and minimized the energy of water molecule (the first 1500 cycles were performed using the fastest descent method, and the total cycles were set at 5000 times). Then, the Cyt c was unrestricted, and minimized the energy of the whole system (the first 2000 cycles were performed using the fastest descent method, and the total cycles were also set at 5000 times). (2) System balance. Temperature and pressure equilibrium processes were based on the Langevin temperature control method and the Isotropic Berendsen pressure control method, respectively. Both of the balance times were set at 100 ps. (3) Unlimited free dynamic simulation. The temperature and pressure control methods were the same as the previous mention. The truncation distance of van der Waal energy or short-range electrostatic energy was set at 10 Å, and the Particle-Mesh-Ewald (PME) method was used to calculate the long-range electrostatic energy. The dynamic simulation time was at least 100 ns, ensuing the Cyt c@HOFs-101 system to reach an equilibrium state.

The $H_2O_2$ binding effect with the heme of native Cyt c and Cyt c@HOF-101 nanosystem was further evaluated by molecular dynamics simulation. Taking native or HOF-101-encapsulated Cyt c as the geometric center, a 1 nm cubic hydrogen peroxide $H_2O_2$ box was added, and $Cl^-$ was added to make the Cyt c electrically neutral. In the overall simulation process, to maintain the structural stability of Cyt c and allow the Cyt c to move flexibly in a certain range, the limiting force of 8370 kJ/mol was added. The molecular dynamics simulation procedure was as same as the steps mentioned above.

### Statistics and reproducibility

The statistic analysis was performed in GraphPad Prism software (version 5.0.1) and Microsoft Excel (version 2016). For electronic and optical microscopy data in the main text (Fig. 2b, c, e) and Supplementary Information (Supplementary Figs. 8a, b; 9a–d; 10; 17b; 18b; 19b, c; 29; 34b), more than three repeats in each experiment were carried out independently with similar results.

### Reporting summary

Further information on research design is available in the Nature Research Reporting Summary linked to this article.

## Data availability

All data supporting this study and its findings are available within the article and its Supplementary Information or from the corresponding authors upon request. The Cyt c structure used herein is available in the PDB database under accession code 6k9i. The structure of GroEL-GroES chaperonin used herein is available in the PDB database under accession code 1pf9. The CAT structure used herein is available in the PDB database under accession code 7di8.

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

## Acknowledgements
We thanks Dr. Xiaomin Ma from Southern University of Science and Technology for the help of cryo-EM imaging. The EPR experiment was performed on the Steady High Magnetic Field Facilities, High Magnetic Field Laboratory, CAS. We acknowledge financial support from projects of the National Natural Science Foundation of China (22174164, G.C.; 22104159, Siming Huang; 21904146, G.C.; 22036003, G.O.; 21737006, G.O.; 22076222, F.Z.), Natural Science Foundation of Guangdong Province (2020A1515010824, G.C.; 2019A1515011722, Siming Huang), Guangzhou Science and Technology Planning Project (202206010074, F.Z.) and Guangdong Provincial Key R&D Programme (2020B1111350002, F.Z.).

## Author contributions
G.C. conceived the idea, designed the experiments, wrote the manuscript and provided financial support. L.T. and G.C. performed material synthesis and characterization. Siming Huang helped with the material synthesis, characterization, data analysis and provided financial support. Shuyao Huang helped with the characterization and data analysis. F.Z. participated in the discussions and provided financial support. G.O. supervised the experiments, wrote the manuscript and provided financial support.

## Competing interests
The authors declare no competing interests.
