## [Peer Review File · Nature Communications]

Hydrogen-bonded organic framework biomimetic entrapment allows enzyme with non-native biocatalytic activityREVIEWER COMMENTS

Reviewer #1 (Remarks to the Author):

The manuscript by Chen et al. reports the conformational change of Cyt c after the immobilization into a HOF (Cyt c@EHN). The immobilized Cyt c shows CAT-like activity, and this was associated to the protein conformational change. The immobilization of enzymes in HOFs is an interesting topic and the manuscript shows an interesting fundamental study about the effect of a specific HOF (HOF-101) on one enzyme (Cyt c). The TEM measurements are outstanding; however, there are major aspects that require improvements prior to considering the manuscript suitable for Nature Communications.

So far, only a few papers have been published in enzyme-HOF composites. In the current paper, some important studies are overlooked: for example, the first seminal work disclosing enzyme-HOF composites is somehow missing (e.g. 10.1021/jacs.9b06589). More recent ones are not included as well (e.g. 10.1002/anie.202110351, 10.1002/anie.202117345). It would be fair to discuss these papers and compare the results obtained in the current study with prior and relevant literature. This will improve the robustness of the manuscript.

The performances of Cyt c@EHN are compared only to the free Catalase. To fully support the claims related to the improved stability of Cyt c@EHN in harsh environments, the authors should compare Cyt c@EHN also to CAT@EHN. This is an important missing control; thus, it is necessary to evaluate the performances of CAT@EHN vs Cyt c@EHN. Despite the interest in the conformational change of Cyt c within the HOF to obtain CAT-like activity, it should be elucidated if CAT maintains or not its conformation within the HOF and if the CAT@EHN performances are better/worse than Cyt c@EHN. The enzyme is reported to be encapsulated in a “defective cage” (e.g. Line 92-93). However, the presented data are not fully conclusive. The authors did not investigate the possibility of having the enzyme immobilized onto the surface of the HOF particles. At the surface, there could be a similar interaction between free COOH groups and the protein as the one discussed in the text. Cyt c should be adsorbed on the surface of HOF particles and this material compared to the one prepared by mixing in one step Cyt c and HOF precursors. Considering the maximum resolution of CLSM, the current images cannot distinguish between surface bounded Cyt c or fully encapsulated Cyt c. In addition, trypsin seems to degrade ca 30% of the Cyt c in Cyt c@EHN: this suggests that 30% of the enzyme was exposed to the surface. However, this deactivation lacks a systematic screening of deactivation parameters. Longer exposure to trypsin and higher trypsin concentrations should be tested to verify whether this value is the maximum deactivation achievable or not. I guess a curve showing a plateau would be a possible way to demonstrate this.

The “de novo” synthesis, originally reported by Shieh et al. (10.1021/ja513058h), involves the use of PVP to facilitate the assembly of the MOF around the protein. In the current manuscript, no PVP was employed. The HOF, ZIF-8 or ZIF-90 precursors and the protein were mixed in water. Unless PVP is added the synthesis should be renamed as a one-pot encapsulation method unless the authors have evidence that the biomolecules spontaneously enhance the formation of the composite (e.g. it seems this case from Figure S1). In this case, the process was originally named “biomimetic mineralization” synthesis (10.1038/ncomms8240). Despite the de-novo approach and the biomimetic mineralization method being both one-pot encapsulation methods, there is experimental evidence that shows how the related composites are structurally (i.e. localization of the enzyme in the composite) and functionally different (10.1021/acs.chemrev.0c01029).

Initial rate measurements should include a linear fit in the initial linear range, showing how the initial rate was determined. This is missing in all enzymatic activity graphs. (e.g. Figure S27, S28, S29, S30) and in the methods it should be described how the fitting was performed. One example is Fig.S7, DOI: 10.1002/anie.202117345 .

One point that does not sound convincing is the analogy towards GroEL-GroES is really the best. GroEL-GroES binds unfolded proteins and facilitates refolding, and then releases the protein in its native state. In this case, the HOF is not assisting the (unfolded/incorrectly folded)enzyme to refold in its native structure but is altering its structure to change its function. Similar to the immobilization of lipases:

“Lipases typically possess a hydrophobic face with a lid that covers the active site, presumably to prevent it hydrolysing the non-lipid esters in a cell. This lid opens in a hydrophobic environment, such

as the surface of a lipid droplet, activating the enzyme. By using a hydrophobic support the lipase can be bound while simultaneously locking the lid into the open position.” (taken from <https://doi.org/10/gmgsgc>).

As Nature Communication is an important journal, it would be relevant to ascertain this point.

Then, statements related to the generality of this approach should be avoided (e.g. “the robust artificial H-bonded network can stabilize the refolded structure of an enzyme”) since only one enzyme was tested. The conclusion should reflect the presented data. Alternatively, 3 or more different enzymes should be examined and supportive experimental evidence should be included in this work.

English should be drastically improved. Examples of issues are:

Line 41: spelling: „no-covalent interactions“ -> „non-covalent“

Line 58: spelling: „This inspirits us“ -> „This inspired us“

Line 62: spelling: „possibility of utilizing an exogenous hydrogen-bonded network as chaperone-like function, to modulate the enzyme folding,“ -> „possibility of utilizing an exogenous hydrogen-bonded network with a chaperone-like function, to modulate the enzyme folding,“

Line 70: grammar: “was also opened by the traction of the H-bonded network that favoring the biocatalysis.”

Line 71: spelling: “The designed H-bonded network has long-range ordered channel, it not only stabilized the refolded enzyme,” -> “channels”

Line 74: grammar: “Our EHN@Cyt c could carry out the CAT-like catalytic function in different harsh conditions that the native CAT should be denatured, and displayed significant enhancements on stability and reusability.”

Line 86: spelling: “idea” -> “ideal”

Line 87: grammar: “2) the periodic carboxyl networks may favor for the biointerface with enzyme.” Not clear

Line 237: grammar: “Among the heme enzymes, Cyt c are the only one species that the heme macrocycle are covalently linked to the protein skeleton11.”

Line 239: grammar: “Hence, the heme macrocycle of our Cyt c nanosystem is much stable compared with other heme enzymes, such as CAT (Figure 5a)”

Line 240: spelling: “Besides, the rigid artificial network, with ultrahigh chemical stability (Figure S26), provided a protecting shell towards Cyt c, and the long-range ordered channel (2.0 nm width) permits many catalytic substrates diffusion and products transportation while excluding the macromolecule.”
Unclear sentence.

Line 243: spelling: “Such structural reinforcements allowed our Cyt c@EHN system perform catalytic tasks in harsh environments.” Missing “to perform”

Figure 5e: The graph starts at “1d” and shows an increase in activity (>10 %), this is misleading, it might be better to add the relative activity before storage (which should be 100%) for clarity.

SI: No instrument is given for how the oxygen concentration was measured. Only: “The time-dependent production of oxygen was recorded using a dissolved oxygen analyzer.”

SI: Scheme S1: spelling: “GroEL-CroES” -> “GroEL-GroES”

Reviewer #2 (Remarks to the Author):

This manuscript by Chen, Ouyang reports the encapsulation of cytochrome C (Cyt C) within a hydrogen bonded framework and the resulting modulation of the coordination environment of the heme active site and protection of the enzyme from deleterious conditions. I think the work is very thoroughly conducted and the results are interesting. In particular there is clearly a significant change in the coordination environment of the heme centre enforced by encapsulation, and the stabilities of the resulting composite is impressive. However I am not convinced by some of the authors' major claims based on their (rigorous) data and I am not sure the work is likely to have the required general impact for a journal of this caliber.

Major concerns:

The paper frequently talks about how the HOF acts similarly to biological chaperones by controlling

the folding of an unfolded protein. Several methods are made to the protein "refolding" inside the HOF. However, I do not think there is any evidence for this process. I agree that encapsulation inside the HOF changes the structure of the protein somewhat - but there is no evidence provided that the protein ever unfolds, so the references to "refolding" do not seem justified. Similarly I do not think the references to "chaperones" are valid unless the authors can demonstrate that the HOF can fold an unfolded protein. If the authors could demonstrate this, then this would be a truly remarkable achievement!

The authors claim that encapsulated Cyt c can catalyze the decomposition of peroxide and claim this has not been seen in native Cyt c (line 159). While I am not a biochemist, this claim seems slightly misleading. While it may be that the decomposition of H₂O₂ into H₂O and O₂ has not been specifically observed, peroxidase type activity in Cyt c is well documented, and indeed the authors show in Figure S12 that the activity of the Cyt c inside the HOF is slightly inferior to the parent Cyt c.

I think several key references are missed. This paper represents one of a very small number of papers (4 or 5) that show that HOFs can encapsulate enzymes and protect them, but it does not cite either the first paper to show this (Liang et al. JACS, 2019, 14298) or another high profile example (Tang et al. Angew. Chem. Int. Ed. 2021, 22315). Similarly the concept of "de novo" enzyme encapsulation is used several times. In one of these (line 64/65) no reference is provided; in another (line 89) the authors cite their own work. I think the authors should cite the original "de novo" enzyme encapsulation papers here, or at least a relevant review. Generally I found the referencing to be heavy on biochemistry/biology papers and quite light on relevant chemistry/HOF papers.

Minor suggestions:

In Scheme 1b, the cartoon of the enzyme before and after encapsulation is identical. I found this a bit confusing, as surely the whole point of this image is that the enzyme changes shape in some way?

I think data could generally be given to fewer significant figures in some places. E.g. do the authors really know the extent of enzyme loading to the nearest 0.1% (line 97), or the calculated binding energies to the nearest 0.01 kcal mol⁻¹ (line 223). On the subject of energies, could these be provided in SI units (kJ mol⁻¹) please?

I am not sure that the statement that Cyt c is more stable because the heme macrocycle is covalently linked to the protein skeleton (lines 237-240) is relevant. Surely the "weakest link" is the tendency of the the protein to denature rather than the heme group to fall out?

In the section on measurement of Cyt c content in the SI, the authors conduct an indirect measurement by looking at the amount of enzyme remaining in the supernatant using a Bradford assay. How confident are the authors that this is accurate? What about surface bound enzyme or enzyme removed in washing? Would a more rigorous approach be to remove any surface bound protein with thorough washing and then digest the composite and determine the amount of enzyme this way?

How many replicates were conducted for MD simulations? Certainly for the data in e.g. Figures S20 and S21, I think repeating the experiment a couple more times and comparing the data would be informative.

Reviewer #3 (Remarks to the Author):

See attachment.

Reviewer #4 (Remarks to the Author):

The observation of diminished or enhanced activity by the porous material in an enzyme-porous material composite is common to a number of previous reports (specifically enhanced or diminished enzyme activity), although notably herein, the authors show quite distinct activity for the enzyme/protein before and after encapsulation - cytochrome C demonstrates catalase-like activity following incorporation in the hydrogen bonded framework (HOF). The claim of chaperone-like activity is a large stretch although one can see the parallels; the use of the term chaperone would require the authors to demonstrate that unfolded/denatured cytochrome C can be refolded into an active state (they only demonstrate modification of a folded enzyme). The term chaperone could be invoked in the discussion as a comment of the future possibilities but I do not think the authors demonstrate true chaperone behaviour.

Regardless, the work will be of interest to those in the field as the use of a porous material to dramatically modify the behaviour of an enzyme post-encapsulation has not been noted; stabilisation/inactivation and improved stability to environmental stressors have been reported but this is a new development. The comparison to ZIF-8 and NU-1000 provide compelling evidence. Overall, the data presented supports the conclusions but the specific nature of the interface between the protein and the HOF needs some tempering; yes, there are modifications to spectral data for certain signals of the HOF and protein but as to the exact interface chemistry definitive data is lacking and the authors might like to reconsider their claims.

Additional experiments

1. The authors should probe the structure and activity of cytochrome C post exposure to a 1:9 DMF-water mixture, as used in the synthesis of the biocomposite. This does not seem to have been used as a control and is needed to eliminate the contribution of the synthetic conditions for composite formation.
2. If "chaperone" is used in the main body of the work then the authors need to show true chaperone like behaviour. Denatured cytochrome C needs to be used for the experiments to show the HOF can partially refold the cytochrome C.

The readability of the manuscript needs some work, especially the introduction.

Response to the reviewers' comments

Reviewer #1 (Remarks to the Author):

The manuscript by Chen et al. reports the conformational change of Cyt c after the immobilization into a HOF (Cyt_c@EHN). The immobilized Cyt c shows CAT-like activity, and this was associated to the protein conformational change. The immobilization of enzymes in HOFs is an interesting topic and the manuscript shows an interesting fundamental study about the effect of a specific HOF (HOF-101) on one enzyme (Cyt c). The TEM measurements are outstanding; however, there are major aspects that require improvements prior to considering the manuscript suitable for Nature Communications.

Response: We thank for the reviewer's positive comments, and we have endeavored to address the issues according to the reviewer's comments.

So far, only a few papers have been published in enzyme-HOF composites. In the current paper, some important studies are overlooked: for example, the first seminal work disclosing enzyme-HOF composites is somehow missing (e.g. 10.1021/jacs.9b06589). More recent ones are not included as well (e.g. 10.1002/anie.202110351, 10.1002/anie.202117345). It would be fair to discuss these papers and compare the results obtained in the current study with prior and relevant literature. This will improve the robustness of the manuscript.

Response: We apologize for the negligence towards the previous articles of enzyme-HOF composites, we have deeply acknowledged and discussed these works in the introduction in the revised manuscript. As the reviewer mentioned, Doonan, Falcaro and co-workers reported the first example of in situ encapsulation of enzymes into HOFs in 2019 (*JACS*, 10.1021/jacs.9b06589). They chose a water-stable, amidinium...carboxylate-based HOFs (BioHOF-1, ca. 6.4 Å pore diameter) as the host, because BioHOF-1 can be synthesized in aqueous phase at room temperature using a tetraamidinium and a tetracarboxylate as the building block. A closer research also founded that the surface-charge modification of enzyme could improve the enzyme encapsulation efficiency, attributing to the enhanced electrostatic effect between enzyme and bioHOF-1 precursors (*Angew. Chem. Int. Ed.*

10.1002/anie.202117345). This has motivated subsequent research to encapsulate protein using analogous amidinium...carboxylate HOFs (e.g. 10.1002/anie.202110351). These preliminary studies provide new possibility of spatially immobilizing an enzyme using amidinium...carboxylate HOFs host.

In this work, the enzyme, Cyt c, was encapsulated into the new mesoporous HOF, HOF-101. HOF-101 was constructed by the π -conjugated carboxylate linker (H4TBAPy), and the intermolecular interaction was not relied on amidinium...carboxylate interaction. We found that Cyt c enabled to trigger the nucleation of the HOF-101 around its surface (**Figure S1** and **S3**), rather than simple encapsulation by electrostatic effect as previously reported. This new encapsulation pattern resulted in the high enzyme loading (39 wt % in HOF-101 vs 6.0 ± 0.5 wt % in BioHOF-1). Importantly, the formed net-carboxyl-arranged defective cage enabled to mediate the conformation of the encapsulated Cyt c into a CAT-like species by the H-bonded nano-biointerface, as evidenced by UV-vis diffuse reflection spectroscopy (UV-vis DRS) (**Figure 2a**), EPR spectra (**Figure 2b**), and the MD simulation (**Figure 4**). Such conformation modulation has not been seen in the previous works including the enzyme-bioHOF-1 system. As far as we known, this is the first example showing the CAT-like function activity of Cyt c under external stimulus by synthetic networks. In addition, the stability of CAT-like activity of Cyt c@EHN was significantly improved compared to the free CAT (**Figure 5**), ascribing to the inherently structural stability of heme center in Cyt c scaffold as well as the protection originated from the HOF-101 shell. We have provided these discussion in the revised manuscript.

The performances of Cyt c@EHN are compared only to the free Catalase. To fully support the claims related to the improved stability of Cyt c@EHN in harsh environments, the authors should compare Cyt c@EHN also to CAT@EHN. This is an important missing control; thus, it is necessary to evaluate the performances of CAT@EHN vs Cyt c@EHN. Despite the interest in the conformational change of Cyt c within the HOF to obtain CAT-like activity, it should be elucidated if CAT maintains or not its conformation within the HOF and if the CAT@EHN performances are better/worse than Cyt c@EHN.

Response: Thank you for the reviewer's meaningful comments. As the reviewer's

suggestion, we have synthesized the CAT@EHN using the similar method, and subsequently compared the stability of CAT-like activity between Cyt c@EHN and CAT@EHN. The obtained CAT@EHN possessed high crystallinity, as verified by the PXRD and SEM image (**Figure R1a** and **R1b**). The typical amide I and amide II bands of protein appeared in the FT-IR spectra of CAT@EHN illustrated the successful encapsulation of CAT (**Figure R1c**). In addition, the standard Bradford assay gave a ca. 44 wt% CAT loading in CAT@EHN. We first confirmed that the CAT retained its activity after encapsulation by the HOF-101 (**Figure R2b**). The slight decrease of CAT-like activity of CAT after encapsulation might be caused by the inevitable inhibition on the substrate diffusion process by the HOFs shell. Such activity inhibitions of enzymes after encapsulation were also widely observed in the case of enzyme@MOFs biocomposites (*Nano Lett.* 2020, 20, 9, 6630–6635; *J. Am. Chem. Soc.* 2017, 139, 6530–6533; *Nat. Commun.* 2019, 10, 5165). The UV-vis diffuse reflection spectroscopy also revealed that the typical adsorption bands of CAT heme were well maintained after the encapsulation, indicating the native conformation of heme center was reserved in CAT@EHN (**Figure R2a**). These results suggested that the HOF-101 scaffold could not mediate the conformation of CAT. We added these data in **Figure R1** and **R2** to the revised supplementary information (**Figure S34** and **Figure S35**)

Figure R1. The PXRD (a), SEM image (b) and FT-IR (c) of synthesized CAT@EHN. The standard Bradford assay gave a ca. 44 wt% CAT loading in CAT@EHN.

Figure R2. (a) The UV-vis DRS revealed that the typical adsorption bands of CAT heme were well maintained after the encapsulation; (b) The CAT bioactivity of free CAT and CAT@EHN. The tests were controlled at the identical enzyme dosage (1 $\mu\text{g}/\text{mL}$) in each group.

As shown in **Figure R3**, the stability of CAT was enhanced after encapsulation by HOF-101, suggesting the protecting effect of HOF-101 shell. However, after different condition treatments, the stabilities of CAT-like activity of Cyt c@EHN were superior to CAT@EHN. These results further suggested that the extraordinary stability of our Cyt c@EHN system was partially resulted from the inherently structural robustness of heme center in Cyt c. We have added these data in **Figure R3** to the revised manuscript (**Figure 5**)

Figure R3. (a) The structure of the heme macrocycle of our Cyt c nanosystem base on MD simulation and native CAT. The retained CAT-like activities of native CAT, CAT@EHN and Cyt c@EHN after acid and weakly alkaline exposure for 30 min (b), heating treatments for 30 min (c), and several denaturing reagents (urea, hydrolase and heavy mental ions, etc.) and organic solvent treatments for 30 min (d). (e) The evaluation of the storage stability of native CAT, CAT@EHN and Cyt c@EHN at room temperature in terms of the CAT-like activity variation.

The enzyme is reported to be encapsulated in a “defective cage” (e.g. Line 92-93). However, the presented data are not fully conclusive. The authors did not investigate the

possibility of having the enzyme immobilized onto the surface of the HOF particles. At the surface, there could be a similar interaction between free COOH groups and the protein as the one discussed in the text. Cyt c should be adsorbed on the surface of HOF particles and this material compared to the one prepared by mixing in one step Cyt c and HOF precursors. Considering the maximum resolution of CLSM, the current images cannot distinguish between surface bounded Cyt c or fully encapsulated Cyt c. In addition, trypsin seems to degrade ca 30% of the Cyt c in Cyt c@EHN: this suggests that 30% of the enzyme was exposed to the surface. However, this deactivation lacks a systematic screening of deactivation parameters. Longer exposure to trypsin and higher trypsin concentrations should be tested to verify whether this value is the maximum deactivation achievable or not. I guess a curve showing a plateau would be a possible way to demonstrate this.

Response: Thank you for the reviewer's comments. We agree with the reviewer's point that the CLSM imaging can not exclude the possibility of surface-adsorbed Cyt c, because the limitation of the maximum resolution of CLSM. To demonstrate the Cyt c was indeed encapsulated into, rather than surface-adsorbed onto HOF-101, we have carried out extra experiments. We firstly examined the surface adsorption capacity of HOF-101 towards Cyt c. The relative narrow mesopore in HOF-101 (ca. 2.0 nm) is not insufficient to accommodate bulky Cyt c (the molecular dimensions of Cyt c are ca. 3.2 nm × 2.7 nm × 3.8 nm, **Figure R4a**). Given this, 5 mg Cyt c were dispersed in 10 mL as-synthesized HOF-101 (10 mg) solution. After 15 min stirring (the time for the de novo assembly is also set at 15 min), the Cyt c-adsorbed HOF-101 was collected, and the enzymes surface-adsorbed within HOF-101 was evaluated based on the concentration change in the supernatants before and after adsorption. The UV spectra of the collected supernatants showed that almost no Cyt c was adsorbed by the HOF-101 (**Figure R4b**). In addition, the apparent color of the Cyt c-adsorbed HOF-101 was consistent with the pure HOF-101, while the color of the Cyt c@EHN changed from yellow to brown, because of the Cyt c incorporation (**Figure R4c**). Furthermore, the FT-IR data also implied that no proteins characteristic peak was recorded in HOF-101 after Cyt c adsorption process (**Figure R4d**). On the contrary, obvious proteins amide I (1700-1610 cm^{-1}) and amide II (1595-1480 cm^{-1}) bands were appeared in Cyt c@EHN. In addition, the significant decrease of N_2 adsorption amounts in

Cyt c@EHN compared with that in HOF-101 also supported the internalization of Cyt c (Figure R5). These results clearly suggested that almost all of Cyt c was encapsulated into, rather surface-adsorbed onto HOF-101. We added these data in Figure R4 and R5 to the revised supplementary information (Figure S3 and Figure S4)

Figure R4. (a) The molecular dimensions of Cyt c based on the PyMOL Molecular Graphics System (Version 2.4.0), and the mesopore structure of MHOFs; (b) The UV spectra and of the collected supernatants before and after adsorption experiment; The apparent color (c) and FT-IR spectra (d) of Cyt c@EHN, HOF-101 and Cyt c-adsorbed HOF-101, respectively.

Figure R5. The Nitrogen adsorption/desorption isotherms of intact HOF-101 and Cyt c@EHN. The N₂ adsorption amount of Cyt c@EHN was significantly lower than that of the intact HOF-101, suggesting that the porosity of Cyt c@EHN was occupied by Cyt c.

As the surface-adsorption experiment has confirmed that almost no Cyt c was adsorbed onto HOF-101, the inhibition on activity of Cyt c@EHN after trypsin treatment (ca. 25 % decrease) should not be the result of the degradation of surface-adsorbed Cyt c, as also evidenced by the fact that no Fe ions was detectable by ICP-MS in the supernatant after trypsin treatment. In addition, the structure of Cyt c@EHN was intact after trypsin treatment (**Figure R6a**). We speculated that such slight inhibition on activity might be resulted from the blockage of pore by trypsin adsorption that limited the transfer mass, because the trypsin concentration used was as high as 5 mg/mL. In addition, the curve by plotting the bioactivity versus various exposure duration showed that ca. 25 % decrease is the maximum deactivation (**Figure R6b**).

Figure R6. (a) SEM image of Cyt c@EHN after 5 mg/mL trypsin treatment for 120 min. (b) The curve by plotting the retained bioactivity of Cyt c@EHN versus various exposure duration in 5 mg/mL trypsin.

The “de novo” synthesis, originally reported by Shieh et al. (10.1021/ja513058h), involves the use of PVP to facilitate the assembly of the MOF around the protein. In the current manuscript, no PVP was employed. The HOF, ZIF-8 or ZIF-90 precursors and the protein were mixed in water. Unless PVP is added the synthesis should be renamed as a one-pot encapsulation method unless the authors have evidence that the biomolecules spontaneously enhance the formation of the composite (e.g. it seems this case from Figure S1). In this case, the process was originally named “biomimetic mineralization” synthesis (10.1038/ncomms8240). Despite the de-novo approach and the biomimetic mineralization

method being both one-pot encapsulation methods, there is experimental evidence that shows how the related composites are structurally (i.e. localization of the enzyme in the composite) and functionally different (10.1021/acs.chemrev.0c01029).

Response: Thanks for this useful suggestion. As the reviewer mentioned, in our assembling system, Cyt c enabled to trigger the nucleation of the HOF-101, and thus accelerated the formation of the composite (**Figure S1**), we agree with the reviewer's view that this process should be termed as "biomimetic mineralization". We have corrected this de novo synthesis method into biomimetic mineralization through the revised manuscript.

Initial rate measurements should include a linear fit in the initial linear range, showing how the initial rate was determined. This is missing in all enzymatic activity graphs. (e.g. Figure S27,S28, S29, S30) and in the methods it should be described how the fitting was performed. One example is Fig.S7, DOI: 10.1002/anie.202117345 .

Response: We sincerely thank the reviewer for his/her careful review. The initial rate was evaluated by the slope of the kinetic curve in the initial phase (from 0 to 40 s), we have stated it in the manuscript text as well as in the corresponding Figure captions (**Figures S30 to S33** in the revised manuscript).

One point that does not sound convincing is the analogy towards GroEL-GroES is really the best. GroEL-GroES binds unfolded proteins and facilitates refolding, and then releases the protein in its native state. In this case, the HOF is not assisting the (unfolded/incorrectly folded)enzyme to refold in its native structure but is altering its structure to change its function. Similar to the immobilization of lipases:

"Lipases typically possess a hydrophobic face with a lid that covers the active site, presumably to prevent it hydrolysing the non-lipid esters in a cell. This lid opens in a hydrophobic environment, such as the surface of a lipid droplet, activating the enzyme. By using a hydrophobic support the lipase can be bound while simultaneously locking the lid into the open position." (taken from <https://doi.org/10/gmmsgc>).

As Nature Communication is an important journal, it would be relevant to ascertain this point.

Response: We completely agree with the reviewer's view, that is, our nanosystem can't assist the (unfolded/incorrectly folded) enzyme to refold in its native structure, yet enables to altering its structure to change its function. According to the reviewer's comment, the claim of GroEL-GroES-like unfolded effect of our nanosystem has been weakened in the revised manuscript, and we only emphasize on its nano-biointerface that allows to mediate the enzyme structure. In addition, we also provided the discussion on the difference between our nanosystem and native GroEL-GroES in the Discussion Section, and we hope the revised version will improve the science and stringency of the manuscript.

Then, statements related to the generality of this approach should be avoided (e.g. "the robust artificial H-bonded network can stabilize the refolded structure of an enzyme") since only one enzyme was tested. The conclusion should reflect the presented data. Alternatively, 3 or more different enzymes should be examined and supportive experimental evidence should be included in this work.

Response: Thanks for the reviewer's meaningful comment. We completely agree with the reviewer's view, and has removed the statements related to the generality of this approach. For example, the statements of "The robust artificial H-bonded network can stabilize the refolded structure of an enzyme" was corrected into "The robust artificial H-bonded network can stabilize the refolded structure of Cyt c."

In fact, we have examined the possibility of our method for mediating the conformation of other heme enzymes, such as horseradish peroxidase (HRP), myoglobin (MB). However, no evidence was found that the confirmations of these two heme enzymes could be altered by the similar process, and no CAT-like activity was observed in HRP@EHN and MB@EHN.

English should be drastically improved. Examples of issues are:

Line 41: spelling: „no-covalent interactions“ -> „non-covalent“

Response: "no-covalent interactions" has been corrected into "non-covalent interactions"

Line 58: spelling: „This inspirits us“ -> „This inspired us“

Response: the misspelled word “inspirit” has been corrected into “inspire” through the revised manuscript.

Line 62: spelling: „possibility of utilizing an exogenous hydrogen-bonded network as chaperone-like function, to modulate the enzyme folding,“ -> „possibility of utilizing an exogenous hydrogen-bonded network with a chaperone-like function, to modulate the enzyme folding,“

Response: Since we have removed the statement of “chaperone-like function” according to other reviewer’s suggestion, this sentence have revised into “possibility of utilizing an exogenous hydrogen-bonded network to modulate the enzyme’s conformation”

Line 70: grammar: “was also opened by the traction of the H-bonded network that favoring the biocatalysis.”

Response: This statement has been corrected into “was also opened by the traction of the H-bonded network that favored the biocatalysis”

Line 71: spelling: “The designed H-bonded network has long-range ordered channel, it not only stabilized the refolded enzyme,“ -> “channels”

Response: “channel” has been corrected into “channels”.

Line 74: grammar: “Our EHN@Cyt c could carry out the CAT-like catalytic function in different harsh conditions that the native CAT should be denatured, and displayed significant enhancements on stability and reusability.”

Response: This statement has been corrected into “Our EHN@Cyt c could carry out the CAT-like catalytic function in different harsh conditions in which free enzyme will be denatured, and displayed significant enhancements on stability and reusability.”

Line 86: spelling: “idea” -> “ideal”

Response: the misspelled word “idea” has been corrected into “ideal” through the revised manuscript.

Line 87: grammar: “2) the periodic carboxyl networks may favor for the biointerface with enzyme.” Not clear

Response: This statement has been corrected into “the periodic carboxyl networks may favor the formation of nano-biointerface because of the potential hydrogen-bonded interaction between the surface residue of enzyme and carboxyl networks”

Line 237: grammar: “Among the heme enzymes, Cyt c are the only one species that the heme macrocycle are covalently linked to the protein skeleton¹¹.”

Response: the statement “Among the heme enzymes, Cyt c are the only one species that the heme macrocycle are covalently linked to the protein skeleton” has been corrected into “Among the heme enzymes, Cyt c is the only one species wherein the heme macrocycle is covalently linked to the protein skeleton”

Line 239: grammar: “Hence, the heme macrocycle of our Cyt c nanosystem is much stable compared with other heme enzymes, such as CAT (Figure 5a)”

Response: This statement has been corrected into “Hence, the heme macrocycle of our Cyt c nanosystem is more structurally stable than other heme enzymes, such as CAT (Figure 5a)”

Line 240: spelling: “Besides, the rigid artificial network, with ultrahigh chemical stability (Figure S26), provided a protecting shell towards Cyt c, and the long-range ordered channel (2.0 nm width) permits many catalytic substrates diffusion and products transportation while excluding the macromolecule.” Unclear sentence.

Response: This statement has been corrected into “Besides, the rigid artificial network, with ultrahigh chemical stability (Figure S29), can protect the inner Cyt c, and its long-range ordered channels (2.0 nm width) ensure the free diffusion and transportation of many catalytic substrates and their products, yet exclude the interferents such as large macromolecules.”

Line 243: spelling: “Such structural reinforcements allowed our Cyt c@EHN system perform catalytic tasks in harsh environments.” Missing “to perform”

Response: This statement has been corrected into “Such structural reinforcements allowed our Cyt c@EHN system to perform catalytic tasks in harsh environments.”

Figure 5e: The graph starts at “1d” and shows an increase in activity (>10 %), this is misleading, it might be better to add the relative activity before storage (which should be 100%) for clarity.

Response: We have added the relative activity before storage (which is 100%) in **Figure 5e** for clarity.

SI: No instrument is given for how the oxygen concentration was measured. Only: “The time-dependent production of oxygen was recorded using a dissolved oxygen analyzer.”

Response: We have provided the detail information of the oxygen measurement in the activity test. Briefly, the Cyt c nanosystem was dispersed into 5 mL PBS (pH=7.4), and the mixed solution was then degassed by bubbling with nitrogen for 20 min. Finally, 50 μ L 100 mM H₂O₂ was rapidly added to initiate the reaction. The catalytic reaction was carried out in an enclosed environment, wherein the probe of the dissolved oxygen analyzer (JBP-607A, Shanghai, China) was immersed into the solution to real-time record the generation of oxygen.

SI: Scheme S1: spelling: “GroEL-CroES” -> “GroEL-GroES”

Response: “GroEL-CroES” has been corrected into “GroEL-GroES” in the revised **Scheme S1**.

Reviewer #2 (Remarks to the Author):

This manuscript by Chen, Ouyang reports the encapsulation of cytochrome C (Cyt C) within a hydrogen bonded framework and the resulting modulation of the coordination

environment of the heme active site and protection of the enzyme from deleterious conditions. I think the work is very thoroughly conducted and the results are interesting. In particular there is clearly a significant change in the coordination environment of the heme centre enforced by encapsulation, and the stabilities of the resulting composite is impressive. However I am not convinced by some of the authors' major claims based on their (rigorous) data and I am not sure the work is likely to have the required general impact for a journal of this caliber.

Response: We sincerely thank the reviewer for his/her positive comments. We have provided more experiments to consolidate our conclusion according to your suggestions, and hope that the revision will improve the robustness of the manuscript.

Major concerns:

The paper frequently talks about how the HOF acts similarly to biological chaperones by controlling the folding of an unfolded protein. Several methods are made to the protein "refolding" inside the HOF. However, I do not think there is any evidence for this process. I agree that encapsulation inside the HOF changes the structure of the protein somewhat - but there is no evidence provided that the protein ever unfolds, so the references to "refolding" do not seem justified. Similarly I do not think the references to "chaperones" are valid unless the authors can demonstrate that the HOF can fold an unfolded protein. If the authors could demonstrate this, then this would be a truly remarkable achievement!

Response: We are sorry for the overestimation of our HOF nanosystem. We completely agree with the reviewer's view that our nanosystem at present enables to change the conformation somewhat, yet, is unable to refold its native structure, as like the chaperones-like function. According to the reviewer's comment, the claim of chaperones-like unfolded effect of our nanosystem has been removed in the revised manuscript, and we only emphasize on its nano-biointerface that allows to mediate or change the Cyt c structure in some degree. Furthermore, we also revised the title into "Hydrogen-bonded network entrapment allows enzyme with new biocatalytic activity".

In addition, we provided the discussion on the difference between our nanosystem and native GroEL-GroES in the Discussion Section, and we hope the revised version will

improve the science and stringency of the manuscript.

The authors claim that encapsulated Cyt c can catalyze the decomposition of peroxide and claim this has not been seen in native Cyt c (line 159). While I am not a biochemist, this claim seems slightly misleading. While it may be that the decomposition of H₂O₂ into H₂O and O₂ has not been specifically observed, peroxidase type activity in Cyt c is well documented, and indeed the authors show in Figure S12 that the activity of the Cyt c inside the HOF is slightly inferior to the parent Cyt c.

Response: We are sorry for this misleading expression. As the reviewer pointed out, the peroxidase type activity in Cyt c has been well documented. We have revised this statement into “Cyt c@EHN was capable of catalyzing the decomposition of H₂O₂ into H₂O and O₂”, and the claim of “this has not been seen in native Cyt c” has been removed in the revised manuscript.

As shown in **Figure S12 (Figure S14** in the revised Supporting information), the inherent peroxidase-like activity of the Cyt c inside the HOF was indeed slightly inferior to the parent Cyt c. This slight decrease of peroxidase-like activity of Cyt c after encapsulation might be caused by the inevitable inhibition on the substrate diffusion process by the HOFs shell. Such activity inhibitions of enzymes after encapsulation were also widely observed in the case of enzyme@MOFs biocomposites (*Nano Lett.* 2020, 20, 9, 6630–6635; *J. Am. Chem. Soc.* 2017, 139, 6530–6533; *Nat. Commun.* 2019, 10, 5165).

I think several key references are missed. This paper represents one of a very small number of papers (4 or 5) that show that HOFs can encapsulate enzymes and protect them, but it does not cite either the first paper to show this (Liang et al. *JACS*, 2019, 14298) or another high profile example (Tang et al. *Angew. Chem. Int. Ed.* 2021, 22315). Similarly the concept of "de novo" enzyme encapsulation is used several times. In one of these (line 64/65) no reference is provided; in another (line 89) the authors cite their own work. I think the authors should cite the original "de novo" enzyme encapsulation papers here, or at least a relevant review. Generally I found the referencing to be heavy on biochemistry/biology papers and quite light on relevant chemistry/HOF papers.

Response: We are sorry for the unbalanced citations. We devoutly appreciate these pioneered works which demonstrate the possibility of encapsulating enzymes using HOFs, and have deeply acknowledged and cited these works (*JACS*, 2019, 14298; *Angew. Chem. Int. Ed.* 2021, 22315; *Angew. Chem. Int. Ed.* 2022, e2021173) in the introduction in the revised manuscript. According to the reviewer's comment, we also cited the representative works of "de novo" enzyme encapsulation in the revised manuscript (*JACS*, 2019, 14298; *Angew. Chem. Int. Ed.* 2021, 22315; *Nano Lett.* 2014, 14, 5761-5765; *Chem. Rev.* 2021, 121, 1077-1129; *Acc. Chem. Res.* 2017, 50, 1423-1432).

In addition, we have cut down the references on biochemistry/biology papers, while enhanced the references on HOF papers (*J. Am. Chem. Soc.* 2011, 133, 14570-14573; *Angew. Chem., Int. Ed.* 2007, 46, 8342-8356; *Chem. Soc. Rev.* 2019, 48, 1362-1389; *J. Am. Chem. Soc.* 2020, 142, 14399-14416; *Acc. Mater. Res.* 2020, 1, 77-87).

Minor suggestions:

In Scheme 1b, the cartoon of the enzyme before and after encapsulation is identical. I found this a bit confusing, as surely the whole point of this image is that the enzyme changes shape in some way?

Response: We are sorry for the unclear presentation of the structural change of enzyme in Scheme 1b, and we have redrawn the Scheme 1b (**Figure R1** in this text) to clarify this structural change.

Figure R1. Mediating the conformation of an enzyme by H-bonded networks encapsulation. (a) Schematic illustration of the protein folding in the typical GroEL-GroES chaperonin system (PDB: 1pf9). (b) Schematic illustration of the designed H-bonded nanotrap for mediating the conformation of an enzyme.

I think data could generally be given to fewer significant figures in some places. E.g. do the authors really know the extent of enzyme loading to the nearest 0.1% (line 97), or the calculated binding energies to the nearest 0.01 kcal mol⁻¹ (line 223). On the subject of energies, could these be provided in SI units (kJ mol⁻¹) please?

Response: The Data have been given to fewer significant figures according to the reviewer's suggestion. The enzyme loading was corrected into ca. 39 %, the units of calculated binding energies have been corrected into kJ mol⁻¹ in the revised manuscript

and Supporting Information, and the significant figures of the values of binding energies were also reduced.

I am not sure that the statement that Cyt c is more stable because the heme macrocycle is covalently linked to the protein skeleton (lines 237-240) is relevant. Surely the "weakest link" is the tendency of the the protein to denature rather than the heme group to fall out?

Response: We agree with the reviewer that this statement is not so precise. We have corrected this statement, and only emphasize that the heme macrocycle of Cyt c is more structurally stable, because it is covalently linked to the protein skeleton. In addition, the statement associated with "heme structure of Cyt c favors the stability of activity" was removed.

In the section on measurement of Cyt c content in the SI, the authors conduct an indirect measurement by looking at the amount of enzyme remaining in the supernatant using a Bradford assay. How confident are the authors that this is accurate? What about surface bound enzyme or enzyme removed in washing? Would a more rigorous approach be to remove any surface bound protein with thorough washing and then digest the composite and determine the amount of enzyme this way?

Response: Thank you for the reviewer's meaningful comments. Indeed, the Cyt c content was evaluated based on the by examining the concentration differences of enzymes in the supernatant before and after assembly via Bradford proteins assay (*Anal. Biochem.* 1976, 72, 248-254). This Bradford proteins assay was widely used for the enzyme loading content in enzyme@MOFs (*J. Chem. Technol. Biotechnol.* 2017, 92, 2583; *Int. J. Biol. Macromol.* 2017, 95, 511; *ACS Sustainable Chem. Eng.* 2019, 7, 19185). In addition, the standard curve of Bradford proteins assay for Cyt c displayed good linear range ($R^2= 0.99$, **Figure R2**), indicating the accurate quantification ability for Cyt c concentration. We added this data in **Figure R2** to the revised supplementary information (**Figure S2a**)

Figure R2. The standard curve for Cyt c quantification using Bradford proteins assay

We further designed other control experiment to examine the surface adsorption capacity of Cyt c onto HOF-101. The relative narrow mesopore in HOF-101 (ca. 2.0 nm) is not insufficient to accommodate bulky Cyt c (the molecular dimension of Cyt c is ca. 3.2 nm × 2.7 nm × 3.8 nm, **Figure R3a**). Given this, 5 mg Cyt c were dispersed in 10 mL as-synthesized HOF-101 (10 mg) solution. After 15 min stirring (the time for the de novo assembly is also set at 15 min), the Cyt c-adsorbed HOF-101 was collected, and the enzymes surface-adsorbed within HOF-101 was evaluated based on the concentration change in the supernatants before and after adsorption. The UV spectra of the collected supernatants showed that almost no Cyt c was adsorbed by the HOF-101 (**Figure R3b**). In addition, the apparent color of the Cyt c-adsorbed HOF-101 was consistent with the pure HOF-101, while the color of the Cyt c@EHN changed from yellow to brown, because of the Cyt c incorporation (**Figure R3c**). Furthermore, the FT-IR data also implied that no proteins characteristic peak was recorded in HOF-101 after Cyt c adsorption process (**Figure R3d**). On the contrary, obvious proteins amide I (1700-1610 cm⁻¹) and amide II (1595-1480 cm⁻¹) bands were appeared in Cyt c@EHN. The aforementioned together demonstrated that the surface adsorption capacity of Cyt c onto HOF-101 was very limited. We added these data in **Figure R3** to the revised supplementary information (**Figure S3**)

Figure R3. (a) The molecular dimensions of Cyt c based on the PyMOL Molecular Graphics System (Version 2.4.0), and the mesopore structure of MHOFS; (b) The UV spectra and of the collected supernatants before and after adsorption experiment; The apparent color (c) and FT-IR spectra (d) of Cyt c@EHN, HOF-101 and Cyt c-adsorbed HOF-101, respectively.

As the reviewer's suggestion, we have washed the as-synthesized Cyt c@EHN with deionized water and ethanol for three times each. The washed Cyt c@EHN was digested by concentrated nitric acid, and then analyzed by ICP-MS (**Figure R4**). Since the heme center of Cyt c has Fe element (one Fe per Cyt c), it enables to determine the amount of Cyt c based on the Fe element in the digested solution by ICP-MS. The ICP-MS analysis gave a Fe content of ca. 0.156 wt% in average in the biocomposites (calculated based on 5 batches of Cyt c@EHN), which was equal to 36.2 wt% Cyt c in average in the biocomposites. This value was in line with the value detected by Bradford proteins assay (ca. 39 wt%). We have provided the relative data in the revised manuscript. We added these data in **Figure R4** to the revised supplementary information (**Figure S2b**), and provided the ICP-MS procedure in the revised supplementary information.

Sample (Cyt c@EHN)	No.1	NO.2	NO.3	NO.4	NO.5	Average
Fe (wt%)	0.153	0.151	0.155	0.178	0.144	0.156
Cyt c (wt%)	35.56	35.00	35.95	41.29	33.36	36.23

Figure R4. The standard curve of Fe ion for Cyt c quantification using ICP-MS, and the calculated Cyt c loading in five batches of Cyt c@EHN.

How many replicates were conducted for MD simulations? Certainly for the data in e.g. Figures S20 and S21, I think repeating the experiment a couple more times and comparing the data would be informative.

Response: Thank for the reviewer's comment. In fact, the Root-mean-square deviations (RMSD) for the Cyt c backbone of free Cyt c and Cyt c@EHN (**Figure S23** and **S24** in the revised manuscript) were replicated for three times in MD simulations, and all of the results showed the same trend.

Reviewer #3 (Remarks to the Author):

The paper by Chen et al. describes the design of a hydrogen-bonded chaperone like network attached to the cytochrome c surface which is embedded in the scaffold. As the enzymatic aspects of the manuscript are outside of my area of expertise, I will specifically comment on the cryo-EM parts of the manuscript.

Response: Thank you for the reviewer's professional comments on the cryo-EM parts, which are very helpful for improving the quality of our manuscript.

In general, the manuscript would benefit as some parts have spelling and grammar errors that make it at time challenging to understand the content.

Response: We are sorry for this mistake, and we have carefully checked and corrected the spelling and grammar errors throughout the revised manuscript.

The data presented in Figure 1C is technically not the IFFT of the area highlighted in Figure 1B, but the IFFT of the FFT of this area. While this might be trivial, I think it would be good to correct this in the text. I am also a bit confused about the annotation of lattice planes and points in the manuscript. The conventions that I am familiar with do not have a [01-1] (Figure 1/S2 and associated text) or (10-3) annotation (Figure S3). Do the authors mean an overbar (1") instead of a – ? If that is the case it should be corrected.

Response: Thanks for this professional comments on TEM images. We have corrected the statement of “the IFFT of the area highlighted in Figure 1B” into the “the IFFT of the FFT of the area highlighted in Figure 1B”.

In addition, The “[01-1] protection” was corrected in to “[01 $\bar{1}$]", and the “(10-3) lattice plane” was corrected into “(10 $\bar{3}$) lattice plane” throughout the text and figures in the revised manuscript.

The methods section for the cryo-EM section would benefit from some additional details. Were the grids directly plunge-frozen in liquid nitrogen without any blotting, as it is common in the field? In addition, it would also be highly uncommon to directly freeze in LN2 and not into liquid ethane, which is used in almost all cryo-EM projects. Was a robot used for freezing or where the grids plunged manually? All these details should be added to the methods as it will help to guide readers that are interested in similar approaches. For the data acquisition it is customary to mention the dose on the camera and how many frames were recorded over a certain time to achieve the final dose. This would be good to add to the methods section as well.

Response: Thanks for the reviewer’s meaningful suggestion. In the cryo-EM experiment, the sample-loaded grids were directly plunge-frozen in liquid nitrogen without any blotting,

and this process was manually operated without using robot.

The data acquisition was operated using SerialEM 3.8. The nominal magnification was set at 350000, with a physical pixel size of 0.34 Å by 0.34 Å. The dose rate on the detector was set at ~15 counts per pixel per second. Each micrograph stack contained 15 frames. The total dose rate was approximately 30 e-/Å² for each micrograph. The motion correction was performed using Motion-Corr2 with 2 × 2 binning, and the non-dose-weighted sum of all frames from each movie was used for all image processing steps. We have provided this details in the cryo-EM method section in the revised manuscript.

Reviewer #4 (Remarks to the Author):

The observation of diminished or enhanced activity by the porous material in an enzyme-porous material composite is common to a number of previous reports (specifically enhanced or diminished enzyme activity), although notably herein, the authors show quite distinct activity for the enzyme/protein before and after encapsulation - cytochrome C demonstrates catalase-like activity following incorporation in the hydrogen bonded framework (HOF). The claim of chaperone-like activity is a large stretch although one can see the parallels; the use of the term chaperone would require the authors to demonstrate that unfolded/denatured cytochrome C can be refolded into an active state (they only demonstrate modification of a folded enzyme). The term chaperone could be invoked in the discussion as a comment of the future possibilities but I do not think the authors demonstrate true chaperone behaviour.

Response: Thank you very much for the reviewer's positive comments. Indeed, as the review mentioned, our HOF nanosystem enabled to modify the conformation of a folded enzyme within its cages, yet, the unfolded/denatured cytochrome C was unable to be refolded into an active state. Viewing from this point, we completely agree with the reviewer's view, that is, the function of our nanosystem at present can not be comparable to the native chaperone. Considering this, the claim of chaperone-like unfolded effect of our nanosystem has been weakened in the revised manuscript, and we only emphasized on its nano-biointerface that allowed to mediate the enzyme structure. In addition, we also

provided the discussion on the difference between our nanosystem and native GroEL-GroES in the Discussion Section, and we hope the revised version will improve the science and stringency of the manuscript.

Regardless, the work will be of interest to those in the field as the use of a porous material to dramatically modify the behaviour of an enzyme post-encapsulation has not been noted; stabilisation/inactivation and improved stability to environmental stressors have been reported but this is a new development. The comparison to ZIF-8 and NU-1000 provide compelling evidence.

Response: Thank you very much for the reviewer's positive comments.

Overall, the data presented supports the conclusions but the specific nature of the interface between the protein and the HOF needs some tempering; yes, there are modifications to spectral data for certain signals of the HOF and protein but as to the exact interface chemistry definitive data is lacking and the authors might like to reconsider their claims.

Response: Thanks for the reviewer's meaningful comments. We recognized that the exact interface between protein and HOFs indeed lacked unambiguous data. In this work, we adopted the FT-IR and solid state NMR (ssNMR) to indirectly demonstrate the interface between protein and HOFs, as evidenced by the shifts of typical FT-IR and ssNMR signal of protein skeleton after encapsulation. So far, the FT-IR and ssNMR are the most widely used method to indirectly explore the biointerface of porous material-confined protein hybrids (*J. Am. Chem. Soc.* 2019, 141, 2348–2355; *JACS Au* 2021, 1, 2172–2181; *Nat. Commun.* 2015, 6, 7240). However, it is still technologically difficult to profile the unambiguous interface data because of the chemical complexities of porous framework as well as the protein structure.

To study the biointerface as much as possible, we carried out the all-atom explicit solvent molecular dynamics (MD) simulations. The simulated results indicated the flexible turn with a mass of NH₂-rich lysine (such as K87、K86、K72、K73, region A in **Figure 4a**), as well as the short helices with lysine (K8 and K27) and other polar residues of glutamine

(Q16), valine (V11) and threonine (T28) of encapsulated Cyt c (region B in **Figure 4a**), strongly interacted with the net-carboxyl arranged in inner wall of the defective cage *via* H-bonded interaction. Such H-bonded interface resulted in the conformation change of the heme center of Cyt c, and this simulated conformation change of heme center was well in agreement with the structural data given by UV-vis diffuse reflection spectroscopy and EPR (the sensitive means for heme coordination study, **Figure 2a** and **2b**).

Therefore, we think that the current data can support the conclusions that our HOF nanosystem enables to modify the conformation of a folded enzyme within its cages by specific interface interaction, even though the exact interface chemistry definitive data is still lacked. In addition, in the discussion section, we have emphasized that the exact interface between protein and HOFs still lacked unambiguous data in our study, and hope that the advanced characterization techniques developed in the future may address this challenge.

Additional experiments

1. The authors should probe the structure and activity of cytochrome C post exposure to a 1:9 DMF-water mixture, as used in the synthesis of the biocomposite. This does not seem to have been used as a control and is needed to eliminate the contribution of the synthetic conditions for composite formation.

Response: Thank you for the reviewer's useful comment. We have carried out this control experiment according to the reviewer's suggestion. The 5 mg Cyt c was directly exposed to 10 mL 1:9 DMF-water mixed solution. After 15 min treatment, the structural information of Cyt c solution was analyzed by UV-vis spectra. As shown in **Figure R1a**, the UV-vis profile of Cyt c was well retained, suggesting that the heme structure of Cyt c could not be changed by this exposure. In addition, no distinct CAT-like activity was recorded in the resultant Cyt c (**Figure R1b**), further verifying that this mild liquid phase could not mediate the conformation of Cyt c. This control experiment also consolidated the conclusion that our HOF nanosystem indeed enabled to modify the conformation of Cyt c within its cages by specific interface interaction. We added these data in **Figure R1** to the revised supplementary information (**Figure S16**)

Figure R1. (a) UV-vis profile of native Cyt c and Cyt c incubated in mixed solution of deionized water and DMF (v/v=9:1; Incubating time: 15 min); (b) The CAT-like catalytic kinetics curves of native Cyt c and Cyt c incubated in mixed solution of water and DMF (v/v=9:1; Incubating time: 15 min). The volume ratio of deionized water and DMF is 9:1, which is consistent with the assembly system.

2. If "chaperone" is used in the main body of the work then the authors need to show true chaperone like behaviour. Denatured cytochrome C needs to be used for the experiments to show the HOF can partially refold the cytochrome C.

Response: Thank you for the reviewer's suggestion. We agree with the review's view, that is, the function of our nanosystem at present could not be comparable to the native chaperone. Our HOFs nanosystem was unable to refold the denatured Cyt c into native ones. Therefore, we have removed the claims of "chaperone-like function" in the revised manuscript. In addition, we also revised the title into "Hydrogen-bonded network entrapment allows enzyme with new biocatalytic activity".

The readability of the manuscript needs some work, especially the introduction.

Response: We are sorry for this mistake. We have carefully checked and corrected the spelling and grammar errors throughout the revised manuscript.

REVIEWER COMMENTS

Reviewer #1 (Remarks to the Author):

The authors replied carefully to all the questions raised by the reviewers. The clarity of the new manuscript is improved and the scientific claims are now more realistic. A few aspects require additional work for the appropriateness of the manuscript for Nature Communication, thus major revisions are needed. Details are reported as follow:

In the abstract, the authors stated “ The resultant H-bonded nanobiointerface mediates the conformation of Cyt c to a catalase (CAT)-like species that previously not been achieved.” Since CAT activity of Cyt c was already reported, the claim should be more specific to the fact that this was not achieved in MOF/HOF materials.

The authors stated “we agree with the reviewer’s view that this process should be termed as “biomimetic mineralization”. We have corrected this de novo synthesis method into biomimetic mineralization through the revised manuscript”. However, the definition of the synthetic procedure as de novo is still present il lines 79, 353 and 211. Thus changes are needed.

Usually, the optimum pH for catalase is between pH 7 and pH 1. In Fig.5, free catalase lost ca 60% of its activity after 30 min at pH8. How is therefore determined the 100% catalytic activity of free catalase? Was it tested in 0.1 mL tris buffer (pH 7.5, 50 mM) as stated in the SI for the peroxidase activity of Cyt c? In general, more experimental details for each test (e.g. exposure to different pHs: concentration? Nature/concentration of the buffer? Was CAT then tested at different pH in the exact same conditions of the HOF materials?) Is the “100% retained activity” referred for all the cases to the free enzyme? All these aspects should be clarified in the text/caption for each graph where the % residual activity is plotted. The procedure should be crystal clear and reproducible by other research groups.

The authors stated in the main text that “The designed H-bonded network has long-range ordered channels, it not only stabilized the encapsulated enzyme, but also accelerated the substrate diffusion.” and repeated this concept several times. However, they then stated “However, the activity of CAT was slightly inhibited after encapsulation (Figure S35b), which might be caused by the infeasible inhibition on the substrate diffusion process by the HOFs shell”. Since the studied substrates are the same, how these two statements could be both valid? All claims that are not experimentally validated by data presented in the paper should be removed.

Reviewer #2 (Remarks to the Author):

The authors have conducted a significant number of additional experiments in response to the comments of the four referees, and have also substantially changed the conceptual framing of the manuscript.

I think these have really improved the work and I am now happy that the scientific conclusions are valid.

In my first review I thought that the manuscript may not be of sufficient novelty/broad interest for Nature Communications, but based on the additional experiments and re-writing I think the manuscript has the broad interest required for this very high level journal.

I suggest that the manuscript be accepted and I do not think any conceptual changes are necessary. There are a few minor English errors, which should be easily fixable at the editing/typesetting stage.

Reviewer #3 (Remarks to the Author):

The authors addressed all my concerns and I recommend the manuscript for publication.

Reviewer #4 (Remarks to the Author):

I am satisfied with the revisions made by the authors, in particular their decision to disassociate themselves from the claim of chaperone-like behaviour. The work has sufficient merit without the ambit claims made in the original abstract, introduction and body of the manuscript.

I do take exception to the additional - unneeded - abbreviation that is introduced in this work, exogenous hydrogen-bonded nanotrap (EHN). Surely the term hydrogen-bonded organic framework (HOF) is sufficient here and will allow the work to be connected to the growing body of work on HOF biocomposites. To distinguish the HOF composites from other work I think the use of HOF-101 as the name of this material is adequate and already used by the authors. It is important also that the authors acknowledge the prior art and making sure that the connection to this work in terms of the name of the class of materials is retained. Also, the HOFs herein are carboxylate not carboxyl, which is the protonated form.

Response to the Reviewers' comments

We thank the reviewers for their constructive and insightful comments, which we have addressed in detail in this point-by-point response. The reviewers' comments are in black with authors' point-by-point responses in blue.

Reviewer #1 (Remarks to the Author):

The authors replied carefully to all the questions raised by the reviewers. The clarity of the new manuscript is improved and the scientific claims are now more realistic. A few aspects require additional work for the appropriateness of the manuscript for Nature Communication, thus major revisions are needed. Details are reported as follow:

Response: We thank for the reviewer's constructive comments, which are very useful for improving the quality of our manuscript.

In the abstract, the authors stated "The resultant H-bonded nanobiointerface mediates the conformation of Cyt c to a catalase (CAT)-like species that previously not been achieved." Since CAT activity of Cyt c was already reported, the claim should be more specific to the fact that this was not achieved in MOF/HOF materials.

Response: We are sorry for this misleading expression. We have corrected the statement of "The resultant H-bonded nanobiointerface mediates the conformation of Cyt c to a catalase (CAT)-like species that previously not been achieved" into "The resultant H-bonded nanobiointerface mediates the conformation of Cyt c to a catalase (CAT)-like species that previously not been achieved in the reported Cyt c-porous organic framework systems"

The authors stated "we agree with the reviewer's view that this process should be termed as "biomimetic mineralization". We have corrected this de novo synthesis method into biomimetic mineralization through the revised manuscript". However, the definition of the synthetic procedure as de novo is still present in lines 79, 353 and 211. Thus changes are needed.

Response: We are sorry for this mistake. In the revised version, the statement of "de novo synthetic procedure" has been corrected into "biomimetic mineralization" throughout the manuscript.

Usually, the optimum pH for catalase is between pH 7 and pH 1. In Fig.5, free catalase lost ca 60% of its activity after 30 min at pH8. How is therefore determined the 100% catalytic activity of free catalase? Was it tested in 0.1 mL tris buffer (pH 7.5, 50 mM) as stated in the SI for the peroxidase activity of Cyt c? In general, more experimental details for each test (e.g. exposure to different pHs: concentration? Nature/concentration of the buffer? Was CAT then tested at different pH in the exact same conditions of the HOF materials?) Is the "100% retained activity" referred for all the cases to the free enzyme? All these aspects should be clarified in the text/caption for each graph where the % residual activity is plotted. The procedure should be crystal clear and reproducible by other research groups.

Response: We apologize for the unclear description on experimental conditions. We have provided more details of the stability tests in the revised manuscript.

Question 1: "How is therefore determined the 100% catalytic activity of free catalase? Was it tested in 0.1 mL tris buffer (pH 7.5, 50 mM) as stated in the SI for the peroxidase activity of Cyt c?"

Reply to Q1: In order to study the effect of pH on bioactivity, the data, shown in Figure 5b, was tested in

deionized water with different pH at room temperature. We did not adopt the buffer solution, because the pH tested ranged from 2 to 8. The traditional buffer solution could not provide such wide buffering range of pH.

Herein, the 100% catalytic activity of free catalase, CAT@HOF or Cyt c@HOF was defined as its bioactivity in pH=7 deionized water at room temperature, respectively. We have provided these details in the revised manuscript.

Question 2: *In general, more experimental details for each test (e.g. exposure to different pHs: concentration? Nature/concentration of the buffer? Was CAT then tested at different pH in the exact same conditions of the HOF materials?)*

Reply to Q2: The experimental details for each test in Figure 5b to 5e have been provided in the Figure captions in the revised manuscript. In order to control the pH in Figure 5b and exclude the potential interaction between metal ions and buffers in Figure 5d, all of the stability tests were carried out in deionized water.

In Figure 5b, enzymes or biocomposites were exposed in different pH deionized water at room temperature for 30 min. The enzyme dosage in different groups was kept at 1 $\mu\text{g}/\text{mL}$. The H_2O_2 used in the activity test was 10 mM.

In Figure 5c, enzymes or biocomposites were dispersed into pH=7 deionized water, and then exposed at different temperatures for 30 min. The enzyme dosage in different groups was kept at 1 $\mu\text{g}/\text{mL}$. The H_2O_2 used in the activity test was 10 mM.

In Figure 5d, enzymes or biocomposites were exposed in different deionized water solution at room temperature for 30 min. Urea: 6 mol/L; Trypsin: 5 mg/mL; All of the metal ions: 10 mmol/L; 80% organic solvent (v/v). The enzyme dosage in different groups was kept at 1 $\mu\text{g}/\text{mL}$. The H_2O_2 used in the activity test was 10 mM.

In Figure 5e, enzymes or biocomposites were dispersed in pH=7 deionized water, and then stood at room temperature for different periods. The enzyme dosage in different groups was kept at 1 $\mu\text{g}/\text{mL}$. The H_2O_2 used in the activity test was 10 mM.

In addition, the CAT tested in different groups from Figure 5b to 5e were in the same conditions as the HOF materials.

We have clarified these experimental details in the captions for each graph.

Question 3: *Is the “100% retained activity” referred for all the cases to the free enzyme?*

Reply to Q3: The “100% retained activity of Cyt c@EHN, CAT@EHN or CAT” was referred for its original bioactivity in pH=7 deionized water at room temperature, respectively. We have clarified this in the text in the revised manuscript.

The authors stated in the main text that “The designed H-bonded network has long-range ordered channels, it not only stabilized the encapsulated enzyme, but also accelerated the substrate diffusion.” and repeated this concept several times. However, they then stated “However, the activity of CAT was slightly inhibited after encapsulation (Figure S35b), which might be caused by the infeasible inhibition on the substrate diffusion process by the HOFs shell”. Since the studied substrates are the same, how these two statements could be both valid? All claims that are not experimentally validated by data presented in the paper should be removed.

Response: Thanks for this insightful comment. As the long-range ordered channels (ca. 2.0 nm) of HOF

were directly identified under the cryo-EM (Figure 1), we think that this large channel enables the diffusion of catalytic substrate into the encapsulated enzyme. The statement of “accelerated the substrate diffusion in Cyt c@HOF-101” is specific to the case in an amorphous carrier. However, compared with the free enzyme, the porous HOFs shell inevitably causes the inhibition on the substrate diffusion process.

We agree with the reviewer’s view that our expression is not rigorous. According to the reviewer’s suggestion, the statement of “accelerated the substrate diffusion” was corrected into “enabled the substrate diffusion” or “ensured the accessibility of biocatalyst” throughout the manuscript.

Reviewer #2 (Remarks to the Author):

The authors have conducted a significant number of additional experiments in response to the comments of the four referees, and have also substantially changed the conceptual framing of the manuscript.

I think these have really improved the work and I am now happy that the scientific conclusions are valid.

In my first review I thought that the manuscript may not be of sufficient novelty/broad interest for Nature Communications, but based on the additional experiments and re-writing I think the manuscript has the broad interest required for this very high level journal.

I suggest that the manuscript be accepted and I do not think any conceptual changes are necessary. There are a few minor English errors, which should be easily fixable at the editing/typesetting stage.

Response: We sincerely thank for the reviewer’s positive comments, and thank again for recommending our work for publication.

Reviewer #3 (Remarks to the Author):

The authors addressed all my concerns and I recommend the manuscript for publication.

Response: We sincerely thank the reviewer for recommending our work for publication.

Reviewer #4 (Remarks to the Author):

I am satisfied with the revisions made by the authors, in particular their decision to disassociate themselves from the claim of chaperone-like behaviour. The work has sufficient merit without the ambit claims made in the original abstract, introduction and body of the manuscript.

Response: We sincerely thank for the reviewer’s positive comments.

I do take exception to the additional - unneeded - abbreviation that is introduced in this work, exogenous hydrogen-bonded nanotrap (EHN). Surely the term hydrogen-bonded organic framework (HOF) is sufficient here and will allow the work to be connected to the growing body of work on HOF biocomposites. To distinguish the HOF composites from other work I think the use of HOF-101 as the name of this material is adequate and already used by the authors. It is important also that the authors acknowledge the prior art and making sure that the connection to this work in terms of the name of the class of materials is retained. Also, the HOFs herein are carboxylate not carboxyl, which is the protonated form.

Response: We thank for the reviewer's constructive suggestion. We have corrected the abbreviation of "hydrogen-bonded nanotrap (EHN)" into "hydrogen-bonded organic framework-101 (HOF-101)" throughout the text and Figures. In addition, the title was also revised into "Hydrogen-bonded organic framework biomimetic entrapment allows enzyme with new biocatalytic activity".

The "carboxyl" was corrected into "carboxylate" throughout the manuscript.

REVIEWER COMMENTS

Reviewer #1 (Remarks to the Author):

The authors replied carefully to all the questions raised by the reviewers. A minor revision, In the context of the disclosure of experimental details, is suggested.

In fact, details related to the procedures used to evaluate the enzymatic activity/stability are now “scattered” across the manuscript and it would be rather difficult for a researcher to reproduce the experiments reported in the manuscript. As we are discussing the final functional properties, these experiments should be easily reproducible.

To improve the clarity of the new information provided by the authors and to ensure the reproducibility of the enzymatic assays from other groups, I suggest summarizing all the details in tables in the SI. For each enzymatic assay, a table should summarize step-by-step the action taken by the authors, the volume, concentration, pH, agitation, etc. of each solution/suspension that is added at each step. In figure S2, the equation of the linear regression should be reported and 5 digits should be reported for each R2.

In Figure S5, the labels are crossed by the lines.

In Figure S12 the significant wavenumbers and the shift should be evidenced with numbers.

Reviewer #4 (Remarks to the Author):

I am satisfied with the changes.

Response to the Reviewers' comments

We thank the reviewers for their constructive and insightful comments, which we have addressed in detail in this point-by-point response. The reviewers' comments are in black with authors' point-by-point responses in blue.

Reviewer #1 (Remarks to the Author):

The authors replied carefully to all the questions raised by the reviewers. A minor revision, in the context of the disclosure of experimental details, is suggested.

Response: We sincerely thank for the reviewer's positive comments.

In fact, details related to the procedures used to evaluate the enzymatic activity/stability are now "scattered" across the manuscript and it would be rather difficult for a researcher to reproduce the experiments reported in the manuscript. As we are discussing the final functional properties, these experiments should be easily reproducible.

To improve the clarity of the new information provided by the authors and to ensure the reproducibility of the enzymatic assays from other groups, I suggest summarizing all the details in tables in the SI. For each enzymatic assay, a table should summarize step-by-step the action taken by the authors, the volume, concentration, pH, agitation, etc. of each solution/suspension that is added at each step.

Response: We thank for the reviewer's constructive comments, which are very useful for improving the readability of our manuscript. We have summarized all the details of CAT-like and peroxidase activity test step-by-step in the Table S2 in the Supplementary Information, and also summarized all details of the stability test step-by-step in Table S3 in the Supplementary Information, respectively.

Table S2. Step-by-step summarization of the details of CAT-like and peroxidase activity tests

Activity test	Step 1	Step 2	Step 3
CAT-like activity	Free Cyt c, Cyt c@ZIF-8, Cyt c@ZIF-90, Cyt c@NU-1000 or Cyt c@HOF-101 was dispersed into 4.5 mL tris buffer (pH 7.5, 50 mM) in a plastic tube, respectively. The final Cyt c concentration in each trials was kept at 0.1 mg/mL.	The mixed solution was then degassed by bubbling with high purity nitrogen for 20 min	500 μ L 100 mM H ₂ O ₂ was rapidly added to initiate the reaction under static state in an enclosed environment. The produced O ₂ was real-time measured by dissolved oxygen analyzer (JBP-607A, Shanghai, China)
peroxidase activity	The free Cyt c or Cyt c@HOF-101 was dispersed into 0.1 mL tris buffer (pH 7.5, 50 mM) in an ultraviolet cuvette. The final Cyt c concentration in each trials was kept at 6.67 μ g/mL.	0.2 mL prepared TMB solution (0.3 mg/mL) was added as the hydrogen donor.	Immediately, 0.2 mL 10 mM H ₂ O ₂ was added to activate the catalytic reaction under static state. The generated oxTMB could be traced at 650 nm by a UV-vis spectrophotometer using a time-scanning mode.

Table S3. Step-by-step summarization of the details of stability tests

Stability test	Step 1	Step 2	Step 3
Stability against pH	Free CAT or biocomposites were exposed in 4.5 mL deionized water with different pH (pH=2, 4, 6, 8) at room temperature for 30 min, respectively. The enzyme dosage in different groups was kept at 0.1 mg/mL.	After exposure, the solution was degassed by bubbling with high purity nitrogen for 20 min	500 μ L 100 mM H ₂ O ₂ was rapidly added to initiate the reaction under static state in an enclosed environment. The O ₂ production rate, which was evaluated by the slope of the kinetic curve in the initial phase from 0 to 40 s, was used for the evaluation of activity change
Stability against heating treatment	Free CAT or biocomposites were dispersed in 4.5 mL pH=7 deionized water in a plastic tube, and then exposed at 60 °C, 80 °C and 100 °C for 30 min, respectively. The enzyme dosage in different groups was kept at 0.1 mg/mL.	After exposure, the solution was degassed by bubbling with high purity nitrogen for 20 min	500 μ L 100 mM H ₂ O ₂ was rapidly added to initiate the reaction under static state in an enclosed environment. The O ₂ production rate, which was evaluated by the slope of the kinetic curve in the initial phase from 0 to 40 s, was used for the evaluation of activity change
Stability against denaturing reagents and organic solvents	Free CAT or biocomposites were exposed in 4.5 mL different solution in a plastic tube at room temperature for 30 min, respectively. Urea: 6 mol/L; Trypsin: 5 mg/mL; All of the metal ions: 10 mmol/L; Organic solvents: 80% (v/v). The enzyme dosage in different groups was kept at 0.1 mg/mL.	After exposure, the solution was degassed by bubbling with high purity nitrogen for 20 min	500 μ L 100 mM H ₂ O ₂ was rapidly added to initiate the reaction under static state in an enclosed environment. The O ₂ production rate, which was evaluated by the slope of the kinetic curve in the initial phase from 0 to 40 s, was used for the evaluation of activity change
Storage stability	Free CAT or biocomposites were dispersed in 4.5 mL pH=7 deionized water in a plastic tube, and then stood at room temperature for 1 d, 2 d, 3d ,4d and 5d, respectively. The enzyme dosage in different groups was kept at 0.1 mg/mL.	After standing for different times, the solution was degassed by bubbling with high purity nitrogen for 20 min	500 μ L 100 mM H ₂ O ₂ was rapidly added to initiate the reaction under static state in an enclosed environment. The O ₂ production rate, which was evaluated by the slope of the kinetic curve in the initial phase from 0 to 40 s, was used for the evaluation of activity change

In figure S2, the equation of the linear regression should be reported and 5 digits should be reported for each R².

Response: Thanks for the reviewer's suggestion. We have added the equations of the linear regression and the R² values with 5 digits in the revised Figure S2.

Sample (Cyt c@HOF-101)	No.1	NO.2	NO.3	NO.4	NO.5	Average
Fe (wt%)	0.153	0.151	0.155	0.178	0.144	0.156
Cyt c (wt%)	35.56	35.00	35.95	41.29	33.36	36.23

Figure S2. Cyt c quantification. (a) The standard curve for Cyt c quantification using Bradford proteins assay; (b) The standard curve of Fe ion for Cyt c quantification using ICP-MS, and the calculated Cyt c loading in five batches of Cyt c@HOF-101.

In Figure S5, the labels are crossed by the lines.

Response: We are sorry for this mistake. We have corrected it in the revised Figure S5.

Figure S5. The high crystallinity of Cyt c@HOF-101. The PXRD patterns of simulated HOF-101 and Cyt c@HOF-101. The Cyt c@HOF-101 well inherited the Bragg diffraction peaks of HOF-101, suggesting the high crystallinity of Cyt c@HOF-101.

In Figure S12 the significant wavenumbers and the shift should be evidenced with numbers.

Response: Thanks for the reviewer's suggestion. We have added the values of the wavenumbers in the revised Figure S12.

Figure S12. Insight into the interface interaction between Cyt c and H-bonded cage by FT-IR. The FT-IR spectra of Cyt c@HOF-101, Cyt c and organic linker of HOF-101. The red-shifts of amide I and II bands of Cyt c in Cyt c@HOF-101 further elucidated the interface interaction between Cyt c and H-bonded networks.

Reviewer #4 (Remarks to the Author):

I am satisfied with the changes.

Response: We sincerely thank for the reviewer's positive comments.